# Semi-flat minima and saddle points by embedding neural networks to overparameterization

**Kenji Fukumizu**[†,‡]         **Shoichiro Yamaguchi**[‡]

**Yoh-ichi Mototake**[†]        **Mirai Tanaka**[†]

[†]The Institute of Statistical Mathematics      [‡]Preferred Networks, Inc.
Tachikawa, Tokyo 190-8562, Japan      Chiyoda-ku, Tokyo 100-0004, Japan
{fukumizu, mototake, mirai}@ism.ac.jp      guguchi@preferred.jp

## Abstract

We theoretically study the landscape of the training error for neural networks in overparameterized cases. We consider three basic methods for embedding a network into a wider one with more hidden units, and discuss whether a minimum point of the narrower network gives a minimum or saddle point of the wider one. Our results show that the networks with smooth and ReLU activation have different partially flat landscapes around the embedded point. We also relate these results to a difference of their generalization abilities in overparameterized realization.

## 1 Introduction

Deep neural networks (DNNs) have been applied to many problems with remarkable successes. On the theoretical understanding of DNNs, however, many problems are still unsolved. Among others, local minima are important issues on learning of DNNs; existence of many local minima is naturally expected by its strong nonlinearity, while people also observe that, with a large network and the stochastic gradient descent, training of DNNs may avoid this issue [8, 9]. For a better understanding of learning, it is essential to clarify the landscape of the training error.

This paper focuses on the error landscape in *overparameterized* situations, where the number of units is surplus to realize a function. This naturally occurs when a large network architecture is employed, and has been recently discussed in connection to optimization and generalization of neural networks ([14, 2, 1] to list a few). To formulate overparameterization rigorously, this paper introduces three basic methods, unit replication, inactive units, and inactive propagation, for embedding a network to a network of more units in some layer. We investigate especially the landscape of the training error around the embedded point, when we embed a minimizer of the error for a smaller model.

A relevant topic to this paper is *flat minima* [6, 7], which attract much attention in literature. Such flatness of minima is often observed empirically, and is connected to generalization performance [3, 8]. There are also some works on how to define flatness appropriately and its relations to generalization [15, 17]. Different from these works, this paper shows some embeddings cause *semi-flat* minima, at which a lower dimensional affine subset in the parameter space gives a constant value of error (see Sec. A). We will also discuss difference between smooth activation and Rectified Linear Unit (ReLU); at a semi-flat minimum obtained by embedding a network of zero training error, the ReLU networks have more flat directions. Using PAC-Bayes arguments [11], we relate this to the difference of generalization bounds between ReLU and smooth networks in overparameterized situations.

This paper extends [4], in which the three embedding methods are discussed and some conditions on minimum points are shown. However, the paper is limited to three-layer networks of smooth activation with one-dimensional output, and addition of one hidden unit is discussed. The current paper covers a much more general class of networks including ReLU activation and arbitrary number of layers, and discusses the difference based on the activation functions as well as a link to generalization.

The main contributions of this paper are summarized as follows.

- Three methods of embedding are introduced for the general $J$-layer networks as basic construction of overparameterized realization of a function (Sec. 2).
- For smooth activation, the unit replication method embeds a minimum to a saddle point under some assumptions (Theorem 5).
- It is shown theoretically that, for ReLU activation, a minimum is always embedded as a minimum by the method of inactive units. The surplus parameters correspond to a flat subset of the training error (Theorem 9). The unit replication gives only saddles under mild conditions (Theorem 10).
- When a network attains zero training error, the embedding by inactive units gives semi-flat minima in both activation models. The ReLU networks give flatter minima in the overparameterized realization, which suggests better generalization through the PAC-Bayes bounds (Sec. 5.2).

All the proofs of the technical results are given in Supplements.

## 2  Neural network and its embedding to a wider model

We discuss $J$ layer, fully connected neural networks that have an activation function $\varphi(\boldsymbol{z}; \boldsymbol{w})$, where $\boldsymbol{z}$ is the input to a unit and $\boldsymbol{w}$ is a parameter vector. The output of the $i$-th unit $\mathcal{U}_i^q$ in the $q$-th layer is recursively defined by $z_i^q = \varphi(\boldsymbol{z}^{q-1}; \boldsymbol{w}_i^q)$, where $\boldsymbol{w}_i^q$ is the weight between $\mathcal{U}_i^q$ and the $(q-1)$-th layer. The activation function $\varphi(\boldsymbol{z}; \boldsymbol{w})$ is any nonlinear function, which often takes the form $\varphi(\boldsymbol{w}_{wgt}^T \boldsymbol{z} - w_{bias})$ with $\boldsymbol{w} = (\boldsymbol{w}_{wgt}, w_{bias})$; typical examples are the sigmoidal function $\varphi(\boldsymbol{z}; \boldsymbol{w}) = \tanh(\boldsymbol{w}_{wgt}^T \boldsymbol{z} - w_{bias})$ and ReLU $\varphi(\boldsymbol{z}; \boldsymbol{w}) = \max\{\boldsymbol{w}_{wgt}^T \boldsymbol{z} - w_{bias}, 0\}$. This paper assumes that there is $\boldsymbol{w}^{(0)}$ such that $\varphi(\boldsymbol{x}; \boldsymbol{w}^{(0)}) = 0$ for any $\boldsymbol{x}$. Focusing the $q$-th layer, with size of the other layers fixed, the set of networks having $H$ units in the $q$-th layer is denoted by $\mathcal{N}_H$. With a parameter $\boldsymbol{\theta}^{(H)} = (W_0, \boldsymbol{w}_1, \ldots, \boldsymbol{w}_H, \boldsymbol{v}_1, \ldots, \boldsymbol{v}_H, V_0)$, the function $\boldsymbol{f}_{\boldsymbol{\theta}^{(H)}}^{(H)}$ of $\mathcal{N}_H$ is defined by

$$\boldsymbol{f}_{\boldsymbol{\theta}^{(H)}}^{(H)}(\boldsymbol{x}) := \boldsymbol{f}^{(H)}(\boldsymbol{x}; \boldsymbol{\theta}^{(H)}) = \boldsymbol{\psi}\big(\textstyle\sum_{j=1}^{H} \boldsymbol{v}_j \varphi(\boldsymbol{x}; \boldsymbol{w}_j, W_0); V_0\big), \tag{1}$$

where $\varphi(\boldsymbol{x}; \boldsymbol{w}_j, W_0)$ is the output of $\mathcal{U}_i^q$ with a summarized parameter $W_0$ in the previous layers, and $\boldsymbol{\psi}(\boldsymbol{z}^{q+1}; V_0)$ is all the parts after $\boldsymbol{z}^{q+1}$ with parameter $V_0$. Note that $\boldsymbol{v}_j$ is a connection weight from the unit $\mathcal{U}_j^q$ to the units in the $(q+1)$-th layer (we omit the bias term for simplicity). The number of units in the $(q-1)$-th and $(q+1)$-th layers are denoted by $D$ and $M$, respectively.

Embedding of a network refers to a map associating a *narrower* network in $\mathcal{N}_{H_0}$ ($H_0 < H$) with a network of a specific parameter in a *wider* model $\mathcal{N}_H$ to realize the same function, keeping other layers unchanged. For clarity, we use $(\boldsymbol{\zeta}_i, \boldsymbol{u}_i)$ instead of $(\boldsymbol{v}_j, \boldsymbol{w}_j)$ for the parameter $\boldsymbol{\theta}^{(H_0)}$ of $\mathcal{N}_{H_0}$;

$$\boldsymbol{f}_{\boldsymbol{\theta}^{(H_0)}}^{(H_0)}(\boldsymbol{x}) := \boldsymbol{f}^{(H_0)}(\boldsymbol{x}; \boldsymbol{\theta}^{(H_0)}) = \boldsymbol{\psi}\big(\textstyle\sum_{i=1}^{H_0} \boldsymbol{\zeta}_i \varphi(\boldsymbol{x}; \boldsymbol{u}_i, W_0); V_0\big). \tag{2}$$

We consider minima and stationary points of the *empirical risk* (or *training error*)

$$L_H(\boldsymbol{\theta}^{(H)}) := \textstyle\sum_{\nu=1}^{n} \ell(\boldsymbol{y}_\nu, \boldsymbol{f}^{(H)}(\boldsymbol{x}_\nu; \boldsymbol{\theta}^{(H)})), \tag{3}$$

where $\ell(\boldsymbol{y}, \boldsymbol{f})$ is a loss function to measure the discrepancy between a teacher $\boldsymbol{y}$ and network output $\boldsymbol{f}$, and $(\boldsymbol{x}_1, \boldsymbol{y}_1), \ldots, (\boldsymbol{x}_n, \boldsymbol{y}_n)$ are given training data. Typical examples of $\ell(\boldsymbol{y}, \boldsymbol{f})$ include the square error $\|\boldsymbol{y} - \boldsymbol{f}\|^2/2$ and logistic loss $-y \log f - (1-y) \log(1-f)$ for $y \in \{0, 1\}$ and $f \in (0, 1)$. In the sequel, we assume the second order differentiability of $\ell(\boldsymbol{y}, \boldsymbol{f})$ with respect to $\boldsymbol{f}$ for each $\boldsymbol{y}$.

### 2.1  Three embedding methods of a network

To fomulate overparameterization, we introduce three basic methods for embedding $\boldsymbol{f}_{\boldsymbol{\theta}^{(H_0)}}^{(H_0)}$ into $\mathcal{N}_H$ so that it realizes exactly the same function as $\boldsymbol{f}_{\boldsymbol{\theta}^{(H_0)}}^{(H_0)}$. See Table 1 and Figure 1 for the definitions.

**(I) Unit replication:** We fix a unit, say the $H_0$-th unit $\mathcal{U}_{H_0}^q$, in $\mathcal{N}_{H_0}$, and replicate it. Simply, $\boldsymbol{\theta}^{(H)}$ has $H - H_0 + 1$ copies of $\boldsymbol{u}_{H_0}$, and divides the weight $\boldsymbol{\zeta}_{H_0}$ by $\boldsymbol{v}_{H_0}, \ldots, \boldsymbol{v}_H$, keeping the other

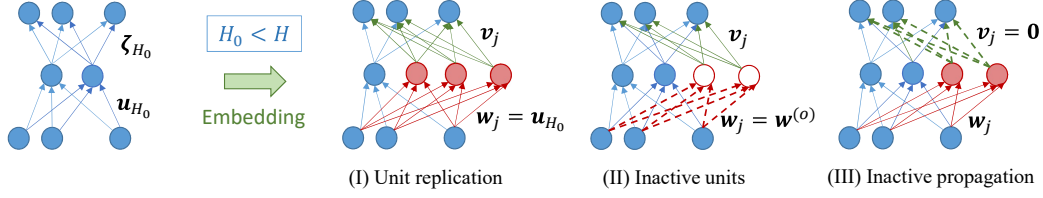

Figure 1: Embedding of a narrower network to a wider one.

| Unit replication $\Pi_{repl}(\boldsymbol{\theta}^{(H_0)})$ | Inactive units $\Pi_{iu}(\boldsymbol{\theta}^{(H_0)})$ | Inactive propagation $\Pi_{ip}(\boldsymbol{\theta}^{(H_0)})$ |
|---|---|---|
| $\boldsymbol{w}_i = \boldsymbol{u}_i \ (1 \leq i \leq H_0 - 1)$ | $\boldsymbol{w}_i = \boldsymbol{u}_i \ (1 \leq i \leq H_0)$ | $\boldsymbol{w}_i = \boldsymbol{u}_i \ (1 \leq i \leq H_0)$ |
| $\boldsymbol{v}_i = \boldsymbol{\zeta}_i \ (1 \leq i \leq H_0 - 1)$ | $\boldsymbol{v}_i = \boldsymbol{\zeta}_i \ (1 \leq i \leq H_0)$ | $\boldsymbol{v}_i = \boldsymbol{\zeta}_i \ (1 \leq i \leq H_0)$ |
| $\boldsymbol{w}_{H_0} = \cdots = \boldsymbol{w}_H = \boldsymbol{u}_{H_0}$ | $\boldsymbol{w}_{H_0+1} = \cdots = \boldsymbol{w}_H = \boldsymbol{w}^{(o)}$ | $\boldsymbol{w}_{H_0+1}, \ldots, \boldsymbol{w}_H$: arbitrary |
| $\boldsymbol{v}_{H_0} + \cdots + \boldsymbol{v}_H = \boldsymbol{\zeta}_{H_0}$ | $\boldsymbol{v}_{H_0+1}, \ldots, \boldsymbol{v}_H$: arbitrary | $\boldsymbol{v}_{H_0+1} = \cdots = \boldsymbol{v}_H = 0$ |

Table 1: Three methods of embedding

parts unchanged. A choice of $\boldsymbol{u}_i \ (1 \leq i \leq H_0)$ to replicate is arbitrary, and a different choice defines a different network. We use $\boldsymbol{u}_{H_0}$ for simplicity. The parameters $\boldsymbol{v}_{H_0}, \ldots, \boldsymbol{v}_H$ consist of an $(H - H_0) \times M$ dimensional affine subspace, denoted by $\Pi_{repl}(\boldsymbol{\theta}^{(H_0)})$, in the parameters for $\mathcal{N}_H$.

**(II) Inactive units:** This embedding uses the special weight $\boldsymbol{w}^{(0)}$ to make the surplus units inactive. The set of parameters is denoted by $\Pi_{iu}(\boldsymbol{\theta}^{(H_0)})$, which is of $(H - H_0) \times M$ dimension.

**(III) Inactive propagation:** This embedding cuts off the weights to the $(q + 1)$-th layer for the surplus part. The weights $\boldsymbol{w}_j$ of the surplus units are arbitrary. The set of parameters is denoted by $\Pi_{ip}(\boldsymbol{\theta}^{(H_0)})$, which is of $(H - H_0) \times D$ dimension.

All the above embeddings give the same function as the narrower network.

**Proposition 1.** *For any $\boldsymbol{\theta}^{(H)} \in \Pi_{repl}(\boldsymbol{\theta}^{(H_0)}) \cup \Pi_{iu}(\boldsymbol{\theta}^{(H_0)}) \cup \Pi_{ip}(\boldsymbol{\theta}^{(H_0)})$, we have $\boldsymbol{f}_{\boldsymbol{\theta}^{(H)}}^{(H)} = \boldsymbol{f}_{\boldsymbol{\theta}^{(H_0)}}^{(H_0)}$.*

It is important to note that a network is not uniquely embedded in a wider model, in contrast to fixed bases models such as the polynomial model. This unidentifiability has been clarified for three-layer networks [10, 16]; in fact, for three layer networks of $\tanh$ activation, [16] shows that the three methods essentially cover all possible embedding. For three-layer networks of 1-dimensional output and smooth activation, [4] shows that this unidentifiable embedding causes minima or saddle points. The current paper extends this result to general networks with ReLU as well as smooth activation.

## 3 Embedding of smooth networks

This section assumes the second order differentiability of $\varphi(\boldsymbol{x}; \boldsymbol{w})$ on $\boldsymbol{w}$. The case of ReLU will be discussed in Sec. 4. Let $\boldsymbol{\theta}_*^{(H_0)}$ be a stationary point of $L_{H_0}$, i.e., $\frac{\partial L_{H_0}(\boldsymbol{\theta}_*^{(H_0)})}{\partial \boldsymbol{\theta}^{(H_0)}} = \boldsymbol{0}$. We are interested in whether the embedding in Sec. 2 also gives a stationary point of $L_H$. More importantly, we wish to know if a minimum of $L_{H_0}$ is embedded to a minimum of $L_H$. A network can be embedded by any combination of the three methods, but we consider their effects separately for simplicity. The definition of minimum, saddle point, and related notions are given by Sec. A.

### 3.1 Stationary properties of embedding

To discuss the stationarity for the case (I) unit replication, we need to restrict $\Pi_{repl}(\boldsymbol{\theta}^{(H_0)})$ to a subset. For $\boldsymbol{\theta}^{(H_0)}$, define $\boldsymbol{\theta}_{\boldsymbol{\lambda}}^{(H)}$ for every $\boldsymbol{\lambda} = (\lambda_{H_0}, \ldots, \lambda_H) \in \mathbb{R}^{H - H_0 + 1}$ with $\sum_{j=H_0}^H \lambda_j = 1$ by

$$\boldsymbol{w}_i = \boldsymbol{u}_i, \quad \boldsymbol{v}_i = \boldsymbol{\zeta}_i \quad (1 \leq i \leq H_0 - 1),$$
$$\boldsymbol{w}_{H_0} = \cdots = \boldsymbol{w}_H = \boldsymbol{u}_{H_0}, \qquad \boldsymbol{v}_j = \lambda_j \boldsymbol{\zeta}_{H_0} \quad (H_0 \leq j \leq H). \tag{4}$$

Obviously, $\boldsymbol{\theta}_{\boldsymbol{\lambda}}^{(H)} \in \Pi_{repl}(\boldsymbol{\theta}^{(H_0)})$ so that $\boldsymbol{f}^{(H)}(\boldsymbol{x}; \boldsymbol{\theta}_{\boldsymbol{\lambda}}^{(H)}) = \boldsymbol{f}^{(H_0)}(\boldsymbol{x}; \boldsymbol{\theta}^{(H_0)})$. The next theorem tells that a stationary point of $\mathcal{N}_{H_0}$ is embedded to an $(H - H_0)$-dimensional stationary subset of $\mathcal{N}_H$.

**Theorem 2.** *Let $\boldsymbol{\theta}_*^{(H_0)}$ be a stationary point of $L_{H_0}$. Then, for any $\boldsymbol{\lambda} = (\lambda_{H_0}, \ldots, \lambda_H)$ with $\sum_{j=H_0}^H \lambda_j = 1$, the point $\boldsymbol{\theta}_{\boldsymbol{\lambda}}^{(H)}$ defined by Eq. (4) is a stationary point of $L_H$.*

The basic idea for the proof is to separate the subset of parameters $(\boldsymbol{v}_{H_0}, \boldsymbol{w}_{H_0}, \ldots, \boldsymbol{v}_H, \boldsymbol{w}_H)$ into a copy of $(\boldsymbol{\zeta}_{H_0}, \boldsymbol{u}_{H_0})$ and the remaining ones, the latter of which do not contribute to change the function $\boldsymbol{f}_{\boldsymbol{\theta}^{(H)}}^{(H)}$ at $\boldsymbol{\theta}_{\boldsymbol{\lambda}}^{(H)}$. We will see this reparameterization in Sec. 3.2 in detail.

It is easy to see that the inactive units or propagations does not generally embed a stationary point to a stationary one (see also Theorems 2 and 4 in [4]). The details will be given in Sec. C.

## 3.2 Embedding of a minimum point in the case of smooth networks

We next consider the embedding $\boldsymbol{\theta}_{\boldsymbol{\lambda}}^{(H)}$ of a mininum point $\boldsymbol{\theta}_*^{(H_0)}$ of $L_{H_0}$. In the sequel, for readability, we discuss three-layer models ($J = 3$) and linear output units. Note however that, for general $J$, the derivatives and Hessian of $L_H$ for the other parameters are exactly the same as those of $L_{H_0}$ for the corresponding parameters. We omit the full description here. The two models are simply given by

$$\mathcal{N}_H : \boldsymbol{f}^{(H)}(\boldsymbol{x}; \boldsymbol{\theta}^{(H)}) = \textstyle\sum_{j=1}^{H} \boldsymbol{v}_j \varphi(\boldsymbol{x}; \boldsymbol{w}_j) \quad \text{and} \quad \mathcal{N}_{H_0} : \boldsymbol{f}^{(H_0)}(\boldsymbol{x}; \boldsymbol{\theta}^{(H_0)}) = \textstyle\sum_{i=1}^{H_0} \boldsymbol{\zeta}_i \varphi(\boldsymbol{x}; \boldsymbol{u}_i).$$ 
(5)

To simplify the Hessian for unit replication, we introduce a new parameterization of $\mathcal{N}_H$. Let $\boldsymbol{\lambda} \in \mathbb{R}^{H-H_0+1}$ be fixed such that $\lambda_{H_0} + \cdots + \lambda_H = 1$ and $\lambda_j \neq 0$. For such $\boldsymbol{\lambda}$, take an $(H - H_0) \times (H - H_0 + 1)$ matrix $A = (\alpha_{cj})$ ($H_0 + 1 \leq c \leq H, H_0 \leq j \leq H$) that satisfies the two conditions:

(A1) $\begin{pmatrix} \mathbf{1}_{H-H_0+1}^T \\ A \end{pmatrix}$ is invertible, where $\mathbf{1}_d = (1, \ldots, 1)^T \in \mathbb{R}^d$,

(A2) $\sum_{j=H_0}^{H} \alpha_{cj} \lambda_j = 0$ for any $H_0 + 1 \leq c \leq H$.

To find such $A$, take $A = (\boldsymbol{a}_{H_0+1}, \ldots, \boldsymbol{a}_H)^T$ so that $\boldsymbol{a}_c^T \boldsymbol{\lambda} = 0$. Then, if $\sum_{c=H_0+1}^{H} s_c \boldsymbol{a}_c = \mathbf{1}_{H-H_0+1}$ for some scalars $s_c$, taking the inner product with $\boldsymbol{\lambda}$ causes a contradiction.

Given such $\boldsymbol{\lambda}$ and $A = (\alpha_{cj})$, define a bijective linear transform from $(\boldsymbol{v}_{H_0}, \ldots, \boldsymbol{v}_H; \boldsymbol{w}_{H_0}, \ldots, \boldsymbol{w}_H)$ to $(\boldsymbol{a}, \boldsymbol{\xi}_{H_0+1}, \ldots, \boldsymbol{\xi}_H; \boldsymbol{b}, \boldsymbol{\eta}_{H_0+1}, \ldots, \boldsymbol{\eta}_H)$ by

$$\boldsymbol{w}_j = \boldsymbol{b} + \textstyle\sum_{c=H_0+1}^{H} \alpha_{cj} \boldsymbol{\eta}_c \quad \text{and} \quad \boldsymbol{v}_j = \lambda_j \boldsymbol{a} + \textstyle\sum_{c=H_0+1}^{H} \lambda_j \alpha_{cj} \boldsymbol{\xi}_c \qquad (H_0 \leq j \leq H).$$ 
(6)

The parameter $\boldsymbol{b}$ serves as the direction that makes all the hidden units behave equally, and $(\boldsymbol{\eta}_j)$ define the remaining $H - 1$ directions that differentiate them. The parameter $\boldsymbol{b}$ thus essentially plays the role of $\boldsymbol{u}_{H_0}$ for $\mathcal{N}_{H_0}$. Also, $\boldsymbol{a}$ works as $\boldsymbol{\zeta}_{H_0}$ when all $\boldsymbol{w}_j$ are equal. The next lemma confirms this role of $(\boldsymbol{a}, \boldsymbol{b})$ and shows that the directions $\boldsymbol{\eta}_c$ and $\boldsymbol{\xi}_c$ do not change the function $\boldsymbol{f}^{(H)}$ at $\boldsymbol{\theta}_{\boldsymbol{\lambda}}^{(H_0)}$.

**Lemma 3.** *Let $\boldsymbol{\theta}^{(H_0)}$ be any parameter of $\mathcal{N}_{H_0}$, and $\boldsymbol{\theta}_{\boldsymbol{\lambda}}^{(H)}$ be its embedding defined by Eq. (4). Then,*

$$\left.\frac{\partial \boldsymbol{f}^{(H)}(\boldsymbol{x}; \boldsymbol{\theta}^{(H)})}{\partial \boldsymbol{b}}\right|_{\boldsymbol{\theta}^{(H)} = \boldsymbol{\theta}_{\boldsymbol{\lambda}}^{(H)}} = \frac{\partial \boldsymbol{f}^{(H_0)}(\boldsymbol{x}; \boldsymbol{\theta}^{(H_0)})}{\partial \boldsymbol{u}_{H_0}}, \qquad \left.\frac{\partial \boldsymbol{f}^{(H)}(\boldsymbol{x}; \boldsymbol{\theta}^{(H)})}{\partial \boldsymbol{\eta}_c}\right|_{\boldsymbol{\theta}^{(H)} = \boldsymbol{\theta}_{\boldsymbol{\lambda}}^{(H)}} = \boldsymbol{0},$$

$$\left.\frac{\partial \boldsymbol{f}^{(H)}(\boldsymbol{x}; \boldsymbol{\theta}^{(H)})}{\partial \boldsymbol{a}}\right|_{\boldsymbol{\theta}^{(H)} = \boldsymbol{\theta}_{\boldsymbol{\lambda}}^{(H)}} = \frac{\partial \boldsymbol{f}^{(H_0)}(\boldsymbol{x}; \boldsymbol{\theta}^{(H_0)})}{\partial \boldsymbol{\zeta}_{H_0}}, \qquad \left.\frac{\partial \boldsymbol{f}^{(H)}(\boldsymbol{x}; \boldsymbol{\theta}^{(H)})}{\partial \boldsymbol{\xi}_c}\right|_{\boldsymbol{\theta}^{(H)} = \boldsymbol{\theta}_{\boldsymbol{\lambda}}^{(H)}} = \boldsymbol{0}.$$
(7)

From Lemma 3, the Hessian takes a simple form:

**Lemma 4.** *Let $\boldsymbol{\lambda}$ and $A$ be as above. Suppose $\boldsymbol{\theta}_*^{(H_0)}$ is a stationary point of $\mathcal{N}_{H_0}$ and $\boldsymbol{\theta}_{\boldsymbol{\lambda}}^{(H)}$ is its embedding defined by Eq. (4). Then, the Hessian matrix of $L_H$ with respect to $\boldsymbol{\omega} = (\boldsymbol{a}, \boldsymbol{b}, \boldsymbol{\xi}_{H_0+1}, \ldots, \boldsymbol{\xi}_H, \boldsymbol{\eta}_{H_0+1}, \ldots, \boldsymbol{\eta}_H)$ at $\boldsymbol{\theta}^{(H)} = \boldsymbol{\theta}_{\boldsymbol{\lambda}}^{(H)}$ is given by*

$$\frac{\partial^2 L_H(\boldsymbol{\theta}_{\boldsymbol{\lambda}}^{(H)})}{\partial \boldsymbol{\omega} \partial \boldsymbol{\omega}} = \begin{array}{c} \\ \boldsymbol{a} \\ \boldsymbol{b} \\ \boldsymbol{\xi}_c \\ \boldsymbol{\eta}_c \end{array} \begin{bmatrix} \overset{\boldsymbol{a}}{\frac{\partial^2 L_{H_0}(\boldsymbol{\theta}_*^{(H_0)})}{\partial \boldsymbol{\zeta}_{H_0} \partial \boldsymbol{\zeta}_{H_0}}} & \overset{\boldsymbol{b}}{\frac{\partial^2 L_{H_0}(\boldsymbol{\theta}_*^{(H_0)})}{\partial \boldsymbol{\zeta}_{H_0} \partial \boldsymbol{u}_{H_0}}} & \overset{\boldsymbol{\xi}_d}{O} & \overset{\boldsymbol{\eta}_d}{O} \\ \frac{\partial^2 L_{H_0}(\boldsymbol{\theta}_*^{(H_0)})}{\partial \boldsymbol{u}_{H_0} \partial \boldsymbol{\zeta}_{H_0}} & \frac{\partial^2 L_{H_0}(\boldsymbol{\theta}_*^{(H_0)})}{\partial \boldsymbol{u}_{H_0} \partial \boldsymbol{u}_{H_0}} & O & O \\ O & O & O & \tilde{F} \\ O & O & \tilde{F}^T & \tilde{G} \end{bmatrix}.$$
(8)

*The lower-right block $\tilde{G} := \left(\frac{\partial^2 L_H(\boldsymbol{\theta}_{\boldsymbol{\lambda}}^{(H)})}{\partial \boldsymbol{\eta}_c \partial \boldsymbol{\eta}_d}\right)_{cd}$, which is a symmetric matrix of $(H - H_0) \times D$ dimension, is given by $\left(A \Lambda A^T\right) \otimes G$ with $\Lambda = \text{Diag}(\lambda_{H_0}, \ldots, \lambda_H)$ and $G := \sum_{\nu=1}^{n} \frac{\partial \ell(\boldsymbol{y}_\nu, \boldsymbol{f}^{(H_0)}(\boldsymbol{x}_\nu; \boldsymbol{\theta}_*^{(H_0)}))}{\partial \boldsymbol{z}} \boldsymbol{\zeta}_{H_0*} \frac{\partial^2 \varphi(\boldsymbol{x}_\nu; \boldsymbol{u}_{H_0*})}{\partial \boldsymbol{u}_{H_0} \partial \boldsymbol{u}_{H_0}}$; and $\tilde{F} := \left(\frac{\partial^2 L_H(\boldsymbol{\theta}_{\boldsymbol{\lambda}}^{(H)})}{\partial \boldsymbol{\xi}_c \partial \boldsymbol{\eta}_d}\right)_{cd}$, which is of size $(H - H_0) \times M$ dimension, is given by $\left(A \Lambda A^T\right) \otimes F$ with $F := \sum_{\nu=1}^{n} \frac{\partial \ell(\boldsymbol{y}_\nu, \boldsymbol{f}^{(H_0)}(\boldsymbol{x}_\nu; \boldsymbol{\theta}_*^{(H_0)}))}{\partial \boldsymbol{z}} \frac{\partial \varphi(\boldsymbol{x}_\nu; \boldsymbol{u}_{H_0*})}{\partial \boldsymbol{u}_{H_0}}$.*

Lemma 4 shows that, with the reparametrization, the Hessian at the embedded stationary point $\boldsymbol{\theta}_{\boldsymbol{\lambda}}^{(H)}$ contains the Hessian of $L_{H_0}$ with $\boldsymbol{a}, \boldsymbol{b}$, and that the cross blocks between $(\boldsymbol{a}, \boldsymbol{b})$ and $(\boldsymbol{\xi}_c, \boldsymbol{\eta}_d)$ are zero. Note that the $\boldsymbol{\xi}$-$\boldsymbol{\xi}$ block is zero, which is important when we prove Theorem 5.

**Theorem 5.** *Consider a three layer network given by Eq. (5). Suppose that the the output dimension $M$ is greater than 1 and $\boldsymbol{\theta}_*^{(H_0)}$ is a minimum of $L_{H_0}$. Let the matrices $G$, $F$ and the parameter $\boldsymbol{\theta}_{\boldsymbol{\lambda}}^{(H)}$ be used in the same meaning as in Lemma 4 (unit replication). Then, if either of the conditions*
*(i) $G$ is positive or negative definite, and $F \neq O$,*
*(ii) $G$ has positive and negative eigenvalues,*
*holds, then for any $\boldsymbol{\lambda}$ with $\sum_{j=H_0}^{H} \lambda_j = 1$ and $\lambda_j \neq 0$, $\boldsymbol{\theta}_{\boldsymbol{\lambda}}^{(H)}$ is a saddle point of $L_H$.*

Theorem 5 is easily proved from Lemma 4. From the form of the lower-right four blocks of Eq. (8), it has positive and negative eigenvalues if $\tilde{G}$ is positive (or negative) definite and $\tilde{F} \neq O$. See Sec. D.3 in Supplements for a complete proof. The assumption $M \geq 2$ is necessary for the condition (i) to happen. In fact, [4] discussed the case of $M = 1$, in which $F = O$ is derived. The paper also gave a sufficient condition that the embedded point $\boldsymbol{\theta}_{\boldsymbol{\lambda}}^{(H)}$ is a local minimum when $G$ is positive (or negative) definite. See Sec. E for more details on the special case of $M = 1$.

Suppose that $\boldsymbol{\theta}_*^{(H_0)}$ attains zero training error. Then, $\boldsymbol{\theta}_{\boldsymbol{\lambda}}^{(H)}$ can never be a saddle point but a global minimum. Therefore, the situation (ii) can never happen. In that case, if $G$ is invertible, it must be positive definite and $F = O$. We will discuss this case further in Sec. 5.1.

# 4 Semi-flat minima by embedding of ReLU networks

This section discusses networks with ReLU. Its special shape causes different results. Let $\phi(t)$ be the ReLU function: $\phi(t) = \max\{t, 0\}$, which is used very often in DNNs to prevent vanishing gradients [12, 5]. The activation is given by $\varphi(\boldsymbol{x}; \boldsymbol{w}) = \phi(\boldsymbol{w}^T \tilde{\boldsymbol{x}})$ with $\boldsymbol{w}^T \tilde{\boldsymbol{x}} := \boldsymbol{w}_{wgt}^T \boldsymbol{x} - w_{bias}$. It is important to note that the ReLU function satisfies positive homogeneity; i.e., $\phi(\alpha t) = \alpha \phi(t)$ for any $\alpha \geq 0$. This causes special properties on $\varphi$, that is, (a) $\varphi(\boldsymbol{x}; r\boldsymbol{w}) = r\varphi(\boldsymbol{x}; \boldsymbol{w})$ for any $r \geq 0$, (b) $\frac{\partial \varphi(\boldsymbol{x};\boldsymbol{w})}{\partial \boldsymbol{w}}\big|_{\boldsymbol{w}=r\boldsymbol{w}_*} = \frac{\partial \varphi(\boldsymbol{x};\boldsymbol{w})}{\partial \boldsymbol{w}}\big|_{\boldsymbol{w}=\boldsymbol{w}_*}$ if $r > 0$, $\boldsymbol{w}^T \tilde{\boldsymbol{x}} \neq 0$, and (c) $\frac{\partial^2 \varphi(\boldsymbol{x};\boldsymbol{w})}{\partial \boldsymbol{w} \partial \boldsymbol{w}} = 0$ if $\boldsymbol{w}^T \tilde{\boldsymbol{x}} \neq 0$.

From the positive homogeneity, effective parameterization needs some normalization of $\boldsymbol{v}_j$ or $\boldsymbol{w}_j$. However, this paper uses the redundant parameterization. In our theoretical arguments, no problem is caused by the redundancy, while it gives additional flat directions in the parameter space.

## 4.1 Embeddings of ReLU networks

Reflecting the above special properties, we introduce modified versions for embeddings of $\boldsymbol{\theta}_*^{(H_0)}$.

**(I)$_R$ Unit replication:** Fix $\mathcal{U}_{H_0}^q$, and take $\boldsymbol{\gamma} = (\gamma_{H_0}, \dots, \gamma_H) \in \mathbb{R}^{H-H_0+1}$ and $\boldsymbol{\beta} = (\beta_{H_0}, \dots, \beta_H)$ such that $\beta_j > 0$ $(H_0 \leq \forall j \leq H)$ and $\sum_{j=H_0}^{H} \gamma_j \beta_j = 1$. Define $\boldsymbol{\theta}_{\boldsymbol{\gamma}, \boldsymbol{\beta}}^{(H)}$ by

$$\boldsymbol{w}_i = \boldsymbol{u}_i, \quad \boldsymbol{v}_i = \boldsymbol{\zeta}_i \quad (1 \leq i \leq H_0 - 1),$$
$$\boldsymbol{w}_j = \beta_j \boldsymbol{u}_{H_0}, \quad \boldsymbol{v}_j = \gamma_j \boldsymbol{\zeta}_{H_0} \quad (H_0 \leq j \leq H). \tag{9}$$

**(II)$_R$ Inactive units:** Define a parameter $\hat{\boldsymbol{\theta}}^{(H)}$ by

$$\boldsymbol{w}_i = \boldsymbol{u}_i, \quad \boldsymbol{v}_i = \boldsymbol{\zeta}_i \quad (1 \leq i \leq H_0), \qquad \boldsymbol{v}_j : \text{ arbitrary} \quad (H_0 + 1 \leq j \leq H)$$
$$\boldsymbol{w}_j \text{ such that } \boldsymbol{w}_j^T \tilde{\boldsymbol{x}}_\nu < 0 \quad (\forall \nu, H_0 + 1 \leq j \leq H). \tag{10}$$

Note that the definition (II)$_R$ is different from the smooth activation case. The last condition is easily satisfied if $w_{bias}$ is large. Note also that $\varphi(\boldsymbol{x}_\nu; \boldsymbol{w}_j) = 0$ for each $\nu$, but $\varphi(\boldsymbol{x}; \boldsymbol{w}_j) \not\equiv 0$ in general. Since a small change of $\boldsymbol{w}_j$ $(H_0 + 1 \leq j \leq H)$ does not alter $\varphi(\boldsymbol{x}_\nu; \boldsymbol{w}_j) = 0$, the function $L_H$ is constant locally on $\boldsymbol{v}_j$ and $\boldsymbol{w}_j$ $(H_0 + 1 \leq j \leq H)$ at $\hat{\boldsymbol{\theta}}^{(H)}$. This is clear difference from the smooth case, where changing $\boldsymbol{w}_j$ from $\boldsymbol{w}^{(0)}$ may cause a different function.

**(III)$_R$ Inactive propagation:** The inactive propagation is exactly the same as the smooth activation case. The embedded point is denoted by $\tilde{\boldsymbol{\theta}}^{(H)}$.

The following proposition is obvious from the definitions.

**Proposition 6.** *For the unit replication and inactive propagation, we have* $\boldsymbol{f}^{(H)}_{\boldsymbol{\theta}^{(H)}_{\gamma,\beta}} = \boldsymbol{f}^{(H)}_{\tilde{\boldsymbol{\theta}}^{(H)}} = \boldsymbol{f}^{(H_0)}_{\boldsymbol{\theta}^{(H_0)}_*}.$

We see that there are some other flat directions in addition to the general cases. In the embedding by inactive units, if the condition $\boldsymbol{w}_j^T \tilde{\boldsymbol{x}}_\nu \leq 0$ is maintained, $L_H$ has the same value. Assume $\|\boldsymbol{x}_\nu\| \leq 1$ without loss of generality, and fix $K > 1$ as a constant. Define $\hat{\boldsymbol{w}}_{j,wgt} = \boldsymbol{0}$ and $\hat{w}_{j,bias} = 2K$ for $H_0 + 1 \leq j \leq H$. From $\boldsymbol{w}_j^T \tilde{\boldsymbol{x}}_\nu \leq \|\boldsymbol{w}_{j,wgt}\| - w_{j,bias} \leq 0$ for $\boldsymbol{w}_j \in B_K := \{\boldsymbol{w}_j \mid \|\boldsymbol{w}_{j,wgt}\| \leq K$ and $K \leq w_{j,bias} \leq 3K\}$ and any $\boldsymbol{v}_j$ ($H_0 + 1 \leq j \leq H$), we have the following result, showing that an $(H - H_0) \times (M + D)$ dimensional affine subset at $\hat{\boldsymbol{\theta}}^{(H)}$ gives the same value at $\boldsymbol{x}_\nu$.

**Proposition 7.** *Assume* $\|\boldsymbol{x}_\nu\| \leq 1$ *($\forall \nu$). If* $(\boldsymbol{v}_i, \boldsymbol{w}_i) = (\boldsymbol{\zeta}_{i*}, \boldsymbol{u}_{i*})$ *($1 \leq i \leq H_0$) and* $(\boldsymbol{v}_j, \boldsymbol{w}_j) \in \mathbb{R}^M \times B_K$ *($H_0 + 1 \leq j \leq H$), we have for any* $\nu = 1, \dots, n$

$$\boldsymbol{f}^{(H)}(\boldsymbol{x}_\nu; \boldsymbol{\theta}^{(H)}) = \boldsymbol{f}^{(H_0)}(\boldsymbol{x}_\nu; \boldsymbol{\theta}^{(H_0)}_*).$$

Next, for the unit replication of ReLU networks, the piecewise linearity of ReLU causes additional flat directions. To see this, for a fixed $(\boldsymbol{\gamma}, \boldsymbol{\beta})$ with $\sum_j \gamma_j \beta_j = 1$, we introduce a parametrization in a similar manner to the smooth case. Let $A = (\alpha_{cj})$ be an $(H - H_0) \times (H - H_0 + 1)$ matrix such that $\sum_{j=H_0}^{H} \alpha_{cj} \gamma_j \beta_j = 0$ ($\forall c$) and $\begin{pmatrix} \mathbf{1}_{H-H_0+1}^T \\ A \end{pmatrix}$ is invertible. Fix such $A$ and define $(\boldsymbol{a}, \boldsymbol{\xi}_{H_0+1}, \dots, \boldsymbol{\xi}_H; \boldsymbol{b}, \boldsymbol{\eta}_{H_0+1}, \dots, \boldsymbol{\eta}_H)$ by Eq. (6). The next proposition shows that a small change of $(\boldsymbol{\eta}_j)_{j=H_0+1}^{H}$ does not alter the value $L_H(\boldsymbol{\theta}^{(H)}) = L_{H_0}(\boldsymbol{\theta}^{(H_0)}_*)$. Let $B_\delta^{\boldsymbol{\eta}}(\boldsymbol{\theta}^{(H)})$ denote the intersection of the ball of radius $\delta > 0$ at $\boldsymbol{\theta}^{(H)}$ and the affine subspace spanned by $\boldsymbol{\eta}_{H_0+1}, \dots, \boldsymbol{\eta}_H$ at $\boldsymbol{\theta}^{(H)}$.

**Proposition 8.** *Let* $\{\boldsymbol{x}_\nu\}_{\nu=1}^n$ *be any data set,* $\boldsymbol{\theta}^{(H_0)}_*$ *be any parameter of the ReLU network* $\mathcal{N}_{H_0}$, *and* $\boldsymbol{\theta}^{(H)}_{\gamma,\beta}$ *be defined by Eq. (9). Assume that* $\boldsymbol{u}_{H_0*}^T \boldsymbol{x}_\nu \neq 0$ *for all* $\nu$. *Then, there is* $\delta > 0$ *such that*

$$\boldsymbol{f}^{(H)}(\boldsymbol{x}_\nu; \boldsymbol{\theta}^{(H)}) = \boldsymbol{f}^{(H_0)}(\boldsymbol{x}_\nu; \boldsymbol{\theta}^{(H_0)}_*) \qquad (\forall \boldsymbol{\theta}^{(H)} \in B_\delta^{\boldsymbol{\eta}}(\boldsymbol{\theta}^{(H)}_{\gamma,\beta}), \forall \nu = 1, \dots, n).$$

See Sec. F.1 for the proof. The situation $\boldsymbol{u}_{H_0*}^T \boldsymbol{x}_\nu \neq 0$ may easily occur in practice (Fig. 2(a)).

### 4.2 Embedding a local minimum of ReLU networks

We first consider the embedding of a minimum by inactive units. Let $\hat{\boldsymbol{\theta}}^{(H)}$ be an embedding of $\boldsymbol{\theta}^{(H_0)}$ by Eq. (10). From Proposition 7, $L_H(\boldsymbol{\theta}^{(H)})$ does not depend on $(\boldsymbol{v}_j, \boldsymbol{w}_j)_{j=H_0+1}^{H}$ around $\hat{\boldsymbol{\theta}}^{(H)}$ but takes the same value as $L_{H_0}(\boldsymbol{\theta}^{(H_0)})$ with $\boldsymbol{\theta}^{(H_0)} = (\boldsymbol{v}_i, \boldsymbol{w}_i)_{i=1}^{H_0}$. We have thus the following theorem.

**Theorem 9.** *Assume that* $\boldsymbol{\theta}^{(H_0)}_*$ *is a minimum of* $L_{H_0}$. *Then, the embedded point* $\hat{\boldsymbol{\theta}}^{(H)}$ *defined by Eq. (10) (inactive units) is a minimum of* $L_H$.

Theorem 9 and Proposition 7 imply that there is an $(H - H_0) \times (M + D)$ dimensional affine subset that gives local minima, and in those directions $L_H$ is flat.

Next, we consider the embedding by unit replication, which needs further restriction on $\boldsymbol{\gamma}$ and $\boldsymbol{\beta}$. Let $\boldsymbol{\theta}^{(H_0)}$ be a parameter of $\mathcal{N}_{H_0}$, and $\boldsymbol{\gamma} = (\gamma_j)_{j=H_0}^{H}$ satisfy $\sum_{j=H_0}^{H} \gamma_j > 0$. Define $\boldsymbol{\theta}^{(H)}_{\gamma}$ by replacing $\boldsymbol{w}_j = \beta_j \boldsymbol{u}_j$ in Eq. (9) with $\boldsymbol{w}_j = \boldsymbol{u}_{H_0} / \sum_{k=H_0}^{H} \gamma_k$ ($H_0 \leq j \leq H$). If we assume $\boldsymbol{u}_{H_0*}^T \boldsymbol{x}_\nu \neq 0$ ($\forall \nu$), the function $L_H$ is differentiable on $\boldsymbol{\eta}_c, \boldsymbol{\xi}_c$, and for the same reason as Theorem 5, the derivatives are zero. By restricting the function on those directions around $\boldsymbol{\theta}^{(H)}_{\gamma}$, from the fact $\frac{\partial^2 \varphi(\boldsymbol{x}_\nu; \boldsymbol{u}_{H_0})}{\partial \boldsymbol{u}_{H_0} \partial \boldsymbol{u}_{H_0}} = 0$, we can see that the Hessian has the form $\begin{pmatrix} O & \tilde{F} \\ \tilde{F}^T & O \end{pmatrix}$, which includes a positive and negative eigenvalue unless $F = O$. This derives the following theorem. (See Sec. F.2 for a complete proof.)

**Theorem 10.** *Suppose that* $\boldsymbol{\theta}^{(H_0)}_*$ *is a minimum point of* $L_{H_0}$. *Assume that* $\boldsymbol{u}_{H_0*}^T \boldsymbol{x}_\nu \neq 0$ *for any* $\nu = 1, \dots, n$, *and that* $F \neq O$ *where* $F$ *is given by Lemma 4. Then, for any* $\boldsymbol{\gamma} \in \mathbb{R}^{H-H_0+1}$ *such that* $\sum_{j=H_0}^{H} \gamma_j > 0$, *the embedded parameter* $\boldsymbol{\theta}^{(H)}_{\gamma}$ *is a saddle point of* $L_H$.

# 5 Discussions

## 5.1 Minimum of zero error

In using a very large network with more parameters than the data size, the training error may reach zero. Assume $\ell(\boldsymbol{y}, \boldsymbol{z}) \geq 0$ and that a narrower model attains $L_{H_0}(\boldsymbol{\theta}_*^{(H_0)}) = 0$ without redundant units, i.e., any deletion of a unit will increase the training error. We investigate overparameterized realization of such a global minimum by embedding in a wider network $\mathcal{N}_H$. Note that by any methods the embedded parameter is a minimum. This causes special local properties on the embedded point.

For simplicity, we assume three-layer networks and $\|\boldsymbol{x}_\nu\| \leq 1 \ (\forall \nu)$. First, consider the unit replication for the smooth activation. As discussed in the last part of Sec. 3.2, the Hessian takes the form

$$\text{Smooth:} \quad \nabla^2 L_H(\boldsymbol{\theta}_{\boldsymbol{\lambda}}^{(H)}) = \begin{matrix} \boldsymbol{\theta}^{(H_0)} \\ \boldsymbol{\eta}_c \\ \boldsymbol{\xi}_c \end{matrix} \begin{bmatrix} \nabla^2 L_{H_0}(\boldsymbol{\theta}_*^{(H_0)}) & O & O \\ O & O & O \\ O & O & \tilde{G} \end{bmatrix}, \tag{11}$$

where $\tilde{G}$ is non-negative definite. It is not difficult to see (Sec. G.2.2) that, in the case of inactive units, the lower-right four blocks take the form $\left( \begin{smallmatrix} O & O \\ O & S \end{smallmatrix} \right)$. The case of inactive propagation is similar.

For ReLU activation, assume $\boldsymbol{\theta}_*^{(H_0)}$ is a differentiable point of $L_{H_0}$ for simplicity. From Proposition 7, the Hessian at the embedding $\hat{\boldsymbol{\theta}}^{(H)}$ by inactive units is given by

$$\text{ReLU:} \quad \nabla^2 L_H(\hat{\boldsymbol{\theta}}^{(H)}) = \begin{matrix} \boldsymbol{\theta}^{(H_0)} & (\boldsymbol{v}_j, \boldsymbol{w}_j) \end{matrix} \begin{bmatrix} \nabla^2 L_{H_0}(\boldsymbol{\theta}_*^{(H_0)}) & O \\ O & O \end{bmatrix}. \tag{12}$$

Similarly to the smooth case, the Hessian for the unit replication $\boldsymbol{\theta}_{\boldsymbol{\gamma}}^{(H)}$ takes the same form as Eq. (12).

## 5.2 Generalization error bounds of embedded networks

Based on the results in Sec. 5.1, here we compare the embedding between ReLU and smooth activation. The results suggest that the ReLU networks can have an advantage in generalization error when zero training error is realized by some type of overparameterized models.

Suppose that the smooth model $\mathcal{N}_{H_{0,s}}$ and ReLU mdoel $\mathcal{N}_{H_{0,r}}$ attain zero training error without redundant units. They are embedded by the method of inactive units into $\mathcal{N}_{H_s}$ and $\mathcal{N}_{H_r}$, respectively, so that $H_s - H_{0,s} = H_r - H_{0,r} (=: E)$ (the same number of surplus units). The dimensionality of the parameters of $\mathcal{N}_{H_{0,s}}$ and $\mathcal{N}_{H_{0,r}}$ are denoted by $d_{sm}^0$ and $d_{rl}^0$, respectively.

The major difference of the local properties in Eqs. (11) and (12) is the existence of matrix $S$ or $\tilde{G}$ in the smooth case. The ReLU network has a flat error surface $L_H$ in both the directions of $\boldsymbol{w}_j$ and $\boldsymbol{v}_j$. In this sense, the embedded minimum is *flatter* in the ReLU network. We relate this difference of semi-flatness to the generalization ability of the networks through the PAC-Bayes bounds, which has been already used for discussing deep learning [13]. Our motivation here is to consider the difference of the activation functions. We give a summary here and defer the details in Sec. G, Supplements.

Let $\mathcal{D}$ be a probability distribution of $(\boldsymbol{x}, \boldsymbol{y})$ and $\mathcal{L}_H(\boldsymbol{\theta}^{(H)}) := E_{\mathcal{D}}[\ell(\boldsymbol{y}, \boldsymbol{f}(\boldsymbol{x}; \boldsymbol{\theta}^{(H)}))]$ be the generalization error (or risk). Training data $(\boldsymbol{x}_1, \boldsymbol{y}_1), \ldots, (\boldsymbol{x}_n, \boldsymbol{y}_n)$ are i.i.d. sample with distribution $\mathcal{D}$. Then, with a trained parameter $\hat{\boldsymbol{\theta}}$, the PAC-Bayes bound tells

$$\mathcal{L}_H(\hat{\boldsymbol{\theta}}) \lesssim \frac{1}{n} L_H(\hat{\boldsymbol{\theta}}) + 2\sqrt{\frac{2(KL(Q\|P) + \ln \frac{2\delta}{n})}{n-1}}, \tag{13}$$

where $P$ is a prior distribution which does *not* depend on the training data, and $Q$ is any distribution such that it distributes on parameters that do not change the value of $L_H$ so much from $L_H(\hat{\boldsymbol{\theta}})$.

We focus on the embedding by inactive units here. See Sec. G.2.3, Supplements, for the other cases. The essential factor of the PAC-Bayes bound is the KL-divergence $KL(Q\|P)$, which is to be small. We use different choices of $P$ and $Q$ for the smooth and ReLU networks (see Sec. G for details). For the smooth networks, $P_{sm}$ is a non-informative normal distribution $N(0, \sigma^2 I_{d_{sm}})$ with

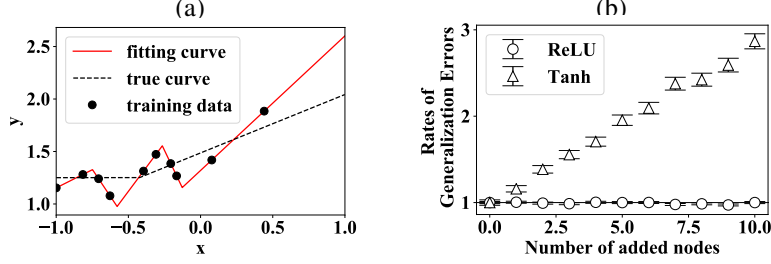

Figure 2: (a) Data and fitting by $\mathcal{N}_5$ with ReLU. (b) Ratio of generalization errors of $\mathcal{N}_H$ and $\mathcal{N}_{H_0}$.

$\sigma \gg 1$, and $Q_{sm}$ is $N(\hat{\boldsymbol{\theta}}_{sm,0}^{(H)}, \tau^2 \mathcal{H}_{sm}^{-1}) \times N(\hat{\boldsymbol{\theta}}_{sm,1}^{(H)}, \sigma^2 I_{d^1}) \times N(\hat{\boldsymbol{\theta}}_{sm,2}^{(H)}, \tau^2 S^{-1})$ with $\tau \ll 1$, where the decomposition corresponds to the components $\boldsymbol{\theta}^{(H_0)}$, $(\boldsymbol{v}_j)_{j=H_0+1}^H$, and $(\boldsymbol{w}_j)_{j=H_0+1}^H$. $\mathcal{H}_{sm} := \nabla^2 L_{H_0}(\boldsymbol{\theta}_{*,sm}^{(H_0)})$ is the Hessian. For ReLU, based on Proposition 7, $P_{rl}$ is given by $N(0, \sigma^2 I_{d_{rl}^0}) \times N(0, \sigma^2 I_{d^1}) \times \text{Unif}_{B_K^E}$, while $Q_{rl}$ is $N(\hat{\boldsymbol{\theta}}_{rl,0}^{(H)}, \tau^2 \mathcal{H}_{rl}^{-1}) \times N(\hat{\boldsymbol{\theta}}_{rl,1}^{(H)}, \sigma^2 I_{d^1}) \times \text{Unif}_{B_K^E}$, where $d^1 = E \times M$ is $\dim(\boldsymbol{v}_j)_{j=H_0+1}^H$. For these choices, the major difference of the bounds is the term

$$d^1 \log\left(\sigma^2/\tau^2\right)$$

in the KL divergence for the smooth model. We can argue that, in realizing perfect fitting to training data with an overparameterized network, the ReLU network achieves a better upper bound of generalization than the smooth network, when the numbers of surplus units are the same.

**Numerical experiments.** We made experiments on the generalization errors of networks with ReLU and $\tanh$ in overparameterization. The input and output dimension is 1. Training data of size 10 are given by $\mathcal{N}_1$ (one hidden unit) for the respective models with additive noise $\varepsilon \sim N(0, 10^{-2})$ in the output. We first trained three-layer networks with each activation to achieve zero training error ($< 10^{-29}$ in squared errors) with minimum number of hidden units ($H_0 = 5$ in both models). See Figure 2(a) for an example of fitting by the ReLU network. We used the method of inactive units for embedding to $\mathcal{N}_H$, and perturb the whole parameters with $N(0, \rho^2)$, where $\rho$ is the $0.01 \times \|\boldsymbol{\theta}_*^{(H_0)}\|$. The code is available in Supplements. Figure 2(b) shows the ratio of the generalization errors (average and standard error for 1000 trials) of $\mathcal{N}_H$ over $\mathcal{N}_{H_0}$ as increasing $H$. We can see that, as more surplus units are added, the generalization errors increase for the $\tanh$ networks, while the ReLU networks do not show such increase. This accords with the theoretical considerations in Sec. 5.2: adding surplus units in $\tanh$ activation makes sharp directions, which degrade the generalization.

### 5.3 Additional remarks

**Regularization.** In training of a large network, one often regularizes parameters based on the norm such as $\ell_2$ or $\ell_1$. Consider, for example, the inactive method of embedding for $\tanh$ or ReLU by setting $\boldsymbol{v}_j = 0$ and $\boldsymbol{w}_j = 0$ ($H_0 + 1 \leq j \leq H$). Then the norm of the embedded parameter is smaller than that of unit replication. This implies that if norm regularization is applied during training, the embedding by inactive units and propagation is to be promoted in overparameterized realization.

**Abundance of semi-flat minima in ReLU networks.** Theorems 9 and 10 discuss three layer models for simplicity, but they can be easily extended to networks of any number of layers. Given a minimum of $L_{H_0}$, it can be embedded to a wider network by making inactive units in any layers. Thus, in a very large (deep and wide) network with overparameterization, there are many affine subsets of parameters to realize the same function, which consist of semi-flat minima of the training error.

## 6 Conclusions

For a better theoretical understanding of the error landscape, this paper has discussed three methods for embedding a network to a wider model, and studied overparameterized realization of a function and its local properties. From the difference of the properties between smooth and ReLU networks, our results suggest that ReLU may have an advantage in realizing zero errors with better generalization. The current analysis reveals some nontrivial geometry of the error landscape, and its implications to dynamics of learning will be within important future works.

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
