[Supplementary Material]

# Supplements to
# "Semi-flat minima and saddle points by embedding neural networks to overparameterization"

## A    Definitions

Let $\Omega$ be an open domain in $\mathbb{R}^d$, and $f : \Omega \to \mathbb{R}$ be a differentiable function. $x_0 \in \Omega$ is called a *mininum* (global minimum) if $f(x) \geq f(x_0)$ for all $x \in \Omega$. $x_0$ is a *local minimum* if there is an open neighborhood $U$ of $x_0$ such that $x_0$ is a minimum $f|_U$ in $U$. A point $x_0 \in \Omega$ is a saddle point if for any open neighborhood of $x_0$ has $y$ and $z$ such that $f(y) > f(x_0)$ and $f(z) < f(x_0)$. Some literature discuss *flat minima* [3, 4, 1, 5], which are observed to have link with generalization performance. In this paper, we introduce *semi-flat minima*, which is defined as an affine subset $V$ of $\Omega$ such that $c = f(x)$ for any $x \in V$ and $f(z) \geq c$ for any $z \in \Omega$.

## B    Proof of Theorem 2

We show a proof using the original parameterization. We can also use the repameteriation introduced in Section 3.2, which may give other insights on the local properties, but we omit it here. See also Figure 3 for the meaning of parameters.

Recall that the gradients of $L_H$ with respect to the parameters can be given by the *back-propagation*, which computes the derivatives with respect to the weight parameters successively from the output layer to the input. For simplicity we use the notation

$$\ell_\nu(\boldsymbol{\theta}^{(H)}) := \ell(\boldsymbol{y}_\nu, \boldsymbol{f}^{(H)}(\boldsymbol{x}_\nu; \boldsymbol{\theta}^{(H)})). \tag{14}$$

Let $\boldsymbol{z}^{k,\nu} = (z_1^{k,\nu}, \dots, z_{H_k}^{k,\nu})^T$ be the input to the $H_k$ units in the $k$-th layer for $\boldsymbol{x}_\nu$, i.e.,

$$z_i^{k,\nu} = \sum_{j=1}^{H_k} w_{ij}^k \phi(z_j^{k-1,\nu}),$$

where $w_{ij}^k$ is the weight parameter connecting from $\mathcal{U}_j^{k-1}$ to $\mathcal{U}_i^k$. Let

$$\delta_i^{k,\nu} := \frac{\partial \ell_\nu(\boldsymbol{\theta}^{(H)})}{\partial z_i^k}.$$

Then, the back-propagation or generalized delta rule [9] computes the derivatives by

$$\delta_j^{k,\nu} = \sum_{i=1}^{H_{k+1}} w_{ij}^{k+1} \delta_i^{k+1} \phi'(z_j^k), \qquad \frac{\partial L_H(\boldsymbol{\theta}^{(H)})}{\partial w_{ij}^k} = \sum_{\nu=1}^n \delta_i^{k,\nu} \phi(z_j^{k-1,\nu}). \tag{15}$$

Now consider the embedding using a unit in the $q$-th layer. Note that the output of any layer except $q$ in $\boldsymbol{f}^{(H)}(\boldsymbol{x}; \boldsymbol{\theta}_\lambda^{(H)})$ is equal to that of $\boldsymbol{f}^{(H_0)}(\boldsymbol{x}; \boldsymbol{\theta}_*^{(H_0)})$, and the backpropagation of the both networks gives exactly the same $\delta_i^{k,\nu}$ to any $\mathcal{U}_{k,i}$ for $k > q$. It follows that

$$\left. \frac{\partial L_H(\boldsymbol{\theta}^{(H)})}{\partial V_0} \right|_{\boldsymbol{\theta}^{(H)} = \boldsymbol{\theta}_\lambda^{(H)}} = \left. \frac{\partial L_{H_0}(\boldsymbol{\theta}^{(H_0)})}{\partial V_0} \right|_{\boldsymbol{\theta}^{(H_0)} = \boldsymbol{\theta}_*^{(H_0)}} = O. \tag{16}$$

The derivatives of $L_{H_0}$ with respect to $\boldsymbol{\zeta}_j$ and $\boldsymbol{u}_j$ $(1 \leq j \leq H_0)$ are given by

$$\frac{\partial L_{H_0}(\boldsymbol{\theta}^{(H_0)})}{\partial \boldsymbol{\zeta}_j} = \sum_{\nu=1}^n \frac{\partial \ell_\nu(\boldsymbol{\theta}^{(H_0)})}{\partial \boldsymbol{z}^{q+1,\nu}} \frac{\partial \boldsymbol{z}^{q+1,\nu}}{\partial \boldsymbol{\zeta}_j} = \sum_{\nu=1}^n \boldsymbol{\delta}^{q+1,\nu} \varphi(\boldsymbol{x}_\nu; \boldsymbol{u}_j, W_0) \tag{17}$$

$$\frac{\partial L_{H_0}(\boldsymbol{\theta}^{(H_0)})}{\partial \boldsymbol{u}_j} = \sum_{\nu=1}^n \frac{\partial \ell_\nu(\boldsymbol{\theta}^{(H_0)})}{\partial \boldsymbol{z}^{q+1,\nu}} \frac{\partial \boldsymbol{z}^{q+1,\nu}}{\partial \boldsymbol{u}_j} = \sum_{\nu=1}^n \boldsymbol{\delta}^{q+1,\nu T} \boldsymbol{\zeta}_j \frac{\partial \varphi(\boldsymbol{x}_\nu; \boldsymbol{u}_j, W_0)}{\partial \boldsymbol{u}_j}, \tag{18}$$

Figure 3: Function of neural networks

where $\boldsymbol{\delta}^{q+1,\nu} = (\delta_1^{q+1,\nu}, \ldots, \delta_M^{q+1,\nu})^T$.

In the same manner, for $1 \leq j \leq H_0 - 1$, the derivatives of $L_H$ with respect to $\boldsymbol{u}_j$ and $\boldsymbol{w}_j$ are given by

$$\frac{\partial L_H(\boldsymbol{\theta}^{(H)})}{\partial \boldsymbol{v}_j} = \sum_{\nu=1}^{n} \frac{\partial \ell_\nu(\boldsymbol{\theta}^{(H)})}{\partial \boldsymbol{z}^{q+1,\nu}} \frac{\partial \boldsymbol{z}^{q+1,\nu}}{\partial \boldsymbol{v}_j} = \sum_{\nu=1}^{n} \boldsymbol{\delta}^{q+1,\nu} \varphi(\boldsymbol{x}_\nu; \boldsymbol{w}_j, W_0) \tag{19}$$

$$\frac{\partial L_H(\boldsymbol{\theta}^{(H)})}{\partial \boldsymbol{w}_j} = \sum_{\nu=1}^{n} \frac{\partial \ell_\nu(\boldsymbol{\theta}^{(H)})}{\partial \boldsymbol{z}^{q+1,\nu}} \frac{\partial \boldsymbol{z}^{q+1,\nu}}{\partial \boldsymbol{w}_j} = \sum_{\nu=1}^{n} \boldsymbol{\delta}^{q+1,\nu T} \boldsymbol{v}_j \frac{\partial \varphi(\boldsymbol{x}_\nu; \boldsymbol{w}_j, W_0)}{\partial \boldsymbol{w}_j}. \tag{20}$$

It is obvious that these derivatives at $\boldsymbol{\theta}^{(H)} = \boldsymbol{\theta}_\lambda^{(H)}$ are equal to those of $L_{H_0}$ at $\boldsymbol{\theta}_*^{(H_0)}$, and thus equal to zero.

For $H_0 \leq j \leq H$, by the definition of $\boldsymbol{\theta}_\lambda^{(H)}$, we have

$$\left.\frac{\partial L_H(\boldsymbol{\theta}^{(H)})}{\partial \boldsymbol{v}_j}\right|_{\boldsymbol{\theta}_\lambda^{(H)}} = \left.\sum_{\nu=1}^{n} \boldsymbol{\delta}^{q+1,\nu} \varphi(\boldsymbol{x}_\nu; \boldsymbol{w}_j, W_0)\right|_{\boldsymbol{\theta}_\lambda^{(H)}} = \sum_{\nu=1}^{n} \boldsymbol{\delta}_*^{q+1,\nu} \varphi(\boldsymbol{x}_\nu; \boldsymbol{u}_{H_0*}, W_{0*}) \tag{21}$$

$$\left.\frac{\partial L_H(\boldsymbol{\theta}^{(H)})}{\partial \boldsymbol{w}_j}\right|_{\boldsymbol{\theta}_\lambda^{(H)}} = \left.\sum_{\nu=1}^{n} \boldsymbol{\delta}^{q+1,\nu T} \boldsymbol{v}_j \frac{\partial \varphi(\boldsymbol{x}_\nu; \boldsymbol{w}_j, W_0)}{\partial \boldsymbol{w}_j}\right|_{\boldsymbol{\theta}_\lambda^{(H)}} = \lambda_j \sum_{\nu=1}^{n} \boldsymbol{\delta}_*^{q+1,\nu T} \boldsymbol{\zeta}_{H_0*} \frac{\partial \varphi(\boldsymbol{x}_\nu; \boldsymbol{u}_{H_0*}, W_{0*})}{\partial \boldsymbol{u}_{H_0}}, \tag{22}$$

which are zero from the stationary condition of $\boldsymbol{\theta}_*^{(H_0)}$. We have also

$$\left.\frac{\partial L_H(\boldsymbol{\theta}^{(H)})}{\partial W_0}\right|_{\boldsymbol{\theta}_\lambda^{(H)}} = \left.\sum_{\nu=1}^{n}\sum_{j=1}^{H} \boldsymbol{\delta}^{q+1,\nu T} \boldsymbol{v}_j \frac{\partial \varphi(\boldsymbol{x}_\nu; \boldsymbol{w}_j, W_0)}{\partial W_0}\right|_{\boldsymbol{\theta}_\lambda^{(H)}}$$

$$= \sum_{\nu=1}^{n} \boldsymbol{\delta}_*^{q+1,\nu T} \sum_{j=1}^{H_0-1} \boldsymbol{\zeta}_{j*} \frac{\partial \varphi(\boldsymbol{x}_\nu; \boldsymbol{u}_{j*}, W_{0*})}{\partial W_0} + \sum_{\nu=1}^{n} \boldsymbol{\delta}_*^{q+1,\nu T} \sum_{j=H_0}^{H} \lambda_j \boldsymbol{\zeta}_{H_0*} \frac{\partial \varphi(\boldsymbol{x}_\nu; \boldsymbol{u}_{H_0*}, W_{0*})}{\partial W_0}$$

$$= \sum_{\nu=1}^{n} \boldsymbol{\delta}_*^{q+1,\nu T} \sum_{j=1}^{H_0} \boldsymbol{\zeta}_{j*} \frac{\partial \varphi(\boldsymbol{x}_\nu; \boldsymbol{u}_{j*}, W_{0*})}{\partial W_0} \tag{23}$$

$$= \left.\frac{\partial L_{H_0}(\boldsymbol{\theta}^{(H_0)})}{\partial W_0}\right|_{\boldsymbol{\theta}^{(H_0)}=\boldsymbol{\theta}_*^{(H_0)}}$$

$$= O, \tag{24}$$

which completes the proof.

## C  Embedding by inactive units and propagation for smooth networks

As in Eqs. (17) through (20), stationary conditions for $L_{H_0}$ give, for $1 \leq i \leq H_0$,

$$\frac{\partial L_{H_0}(\boldsymbol{\theta}^{(H_0)})}{\partial \boldsymbol{\zeta}_i} = \sum_{\nu=1}^{n} \boldsymbol{\delta}^{q+1,\nu} \varphi(\boldsymbol{x}_\nu; \boldsymbol{u}_i, W_0) = \boldsymbol{0}$$

$$\frac{\partial L_{H_0}(\boldsymbol{\theta}^{(H_0)})}{\partial \boldsymbol{u}_j} = \sum_{\nu=1}^{n} \boldsymbol{\delta}^{q+1,\nu T} \boldsymbol{\zeta}_i \frac{\partial \varphi(\boldsymbol{x}_\nu; \boldsymbol{u}_i, W_0)}{\partial \boldsymbol{u}_i} = \boldsymbol{0}. \tag{25}$$

The derivatives of $L_H$ with respect to $\boldsymbol{v}_j$ and $\boldsymbol{w}_j$ are given by

$$\frac{\partial L_H(\boldsymbol{\theta}^{(H)})}{\partial \boldsymbol{v}_j} = \sum_{\nu=1}^{n} \frac{\partial \ell_\nu(\boldsymbol{\theta}^{(H)})}{\partial \boldsymbol{z}^{q+1,\nu}} \frac{\partial \boldsymbol{z}^{q+1,\nu}}{\partial \boldsymbol{v}_j} = \sum_{\nu=1}^{n} \boldsymbol{\delta}^{q+1,\nu} \varphi(\boldsymbol{x}_\nu; \boldsymbol{w}_j, W_0) \tag{26}$$

$$\frac{\partial L_H(\boldsymbol{\theta}^{(H)})}{\partial \boldsymbol{w}_j} = \sum_{\nu=1}^{n} \frac{\partial \ell_\nu(\boldsymbol{\theta}^{(H)})}{\partial \boldsymbol{z}^{q+1,\nu}} \frac{\partial \boldsymbol{z}^{q+1,\nu}}{\partial \boldsymbol{w}_j} = \sum_{\nu=1}^{n} \boldsymbol{\delta}^{q+1,\nu T} \boldsymbol{v}_j \frac{\partial \varphi(\boldsymbol{x}_\nu; \boldsymbol{w}_j, W_0)}{\partial \boldsymbol{w}_j}. \tag{27}$$

In the case of inactive units, $\boldsymbol{v}_j$ for $j \geq H_0 + 1$ is arbitrary and the $\frac{\partial \varphi(\boldsymbol{x}_\nu; \boldsymbol{w}^{(0)}, W_0)}{\partial \boldsymbol{w}_j}$ is not necessarily zero, so that Eq. (25) does not necessarily imply that Eq. (27) is zero. In the case of inactive propagation, $\boldsymbol{w}_j$ is arbitrary for $j \geq H_0 + 1$, which does not mean Eq. (26) is zero in general.

Consider the embedding by making both of units and propagation inactive; i.e.,

$$\boldsymbol{v}_i = \boldsymbol{\zeta}_i \quad (1 \leq i \leq H_0)$$
$$\boldsymbol{w}_i = \boldsymbol{u}_i \quad (1 \leq i \leq H_0)$$
$$\boldsymbol{v}_j = \boldsymbol{0} \quad (H_0 + 1 \leq j \leq H)$$
$$\boldsymbol{w}_j = \boldsymbol{w}^{(0)} \quad (H_0 + 1 \leq j \leq H). \tag{28}$$

Then, for $j \geq H_0 + 1$, we have $\varphi(\boldsymbol{x}; \boldsymbol{w}_j, W_0) = 0$ at $\boldsymbol{w}_j = \boldsymbol{w}^{(0)}$ which means Eq. (26) is zero, and Eq. (27) vanishes from $\boldsymbol{v}_j = \boldsymbol{0}$. Therefore, the stationary point of $L_{H_0}$ is embedded to a stationary point of $L_H$, but there is no flat direction for this stationary point in general.

## D  Proofs of Lemmas 3, 4, and Theorem 5 in Section 3

In the sequel, we repeatedly use the following relations.

$$\frac{\partial}{\partial \boldsymbol{b}} = \sum_{j=H_0}^{H} \frac{\partial}{\partial \boldsymbol{w}_j}, \qquad\qquad \frac{\partial}{\partial \boldsymbol{\eta}_c} = \sum_{j=H_0}^{H} \alpha_{cj} \frac{\partial}{\partial \boldsymbol{w}_j},$$

$$\frac{\partial}{\partial \boldsymbol{a}} = \sum_{j=H_0}^{H} \lambda_j \frac{\partial}{\partial \boldsymbol{v}_j}, \qquad\qquad \frac{\partial}{\partial \boldsymbol{\xi}_c} = \sum_{k=H_0}^{H} \lambda_k \alpha_{ck} \frac{\partial}{\partial \boldsymbol{v}_k}. \tag{29}$$

### D.1  Proof of Lemma 3

It follows from Eq. (29) that

$$\frac{\partial \boldsymbol{f}^{(H)}(\boldsymbol{x}; \boldsymbol{\theta}^{(H)})}{\partial \boldsymbol{b}}\bigg|_{\boldsymbol{\theta}^{(H)}=\boldsymbol{\theta}_\lambda^{(H)}} = \sum_{j=H_0}^{H} \frac{\partial \boldsymbol{f}^{(H)}(\boldsymbol{x}; \boldsymbol{\theta}^{(H)})}{\partial \boldsymbol{w}_j}\bigg|_{\boldsymbol{\theta}^{(H)}=\boldsymbol{\theta}_\lambda^{(H)}}$$

$$= \sum_{j=H_0}^{H} \boldsymbol{v}_j \frac{\partial \varphi(\boldsymbol{x}; \boldsymbol{w}_j)}{\partial \boldsymbol{w}_j}\bigg|_{\boldsymbol{\theta}^{(H)}=\boldsymbol{\theta}_\lambda^{(H)}}$$

$$= \sum_{j=H_0}^{H} \lambda_j \boldsymbol{\zeta}_{H_0} \frac{\partial \varphi(\boldsymbol{x}; \boldsymbol{u}_{H_0})}{\partial \boldsymbol{u}_{H_0}}$$

$$= \frac{\partial \boldsymbol{f}^{(H_0)}(\boldsymbol{x}; \boldsymbol{\theta}_*^{(1)})}{\partial \boldsymbol{u}_{H_0}},$$

since $\sum_j \lambda_j = 1$. Also,

$$\left. \frac{\partial \boldsymbol{f}^{(H)}(\boldsymbol{x}; \boldsymbol{\theta}^{(H)})}{\partial \boldsymbol{\eta}_c} \right|_{\boldsymbol{\theta}^{(H)}=\boldsymbol{\theta}_\lambda^{(H)}} = \sum_{j=H_0}^{H} \alpha_{cj} \left. \frac{\partial \boldsymbol{f}^{(H)}(\boldsymbol{x}; \boldsymbol{\theta}^{(H)})}{\partial \boldsymbol{w}_j} \right|_{\boldsymbol{\theta}^{(H)}=\boldsymbol{\theta}_\lambda^{(H)}}$$

$$= \sum_{j=H_0}^{H} \alpha_{cj} \lambda_j \boldsymbol{\zeta}_{H_0} \frac{\partial \varphi(\boldsymbol{x}; \boldsymbol{u}_{H_0})}{\partial \boldsymbol{u}_{H_0}} = 0,$$

since $\sum_j \alpha_{cj} \lambda_j = 0$ by definition of $A$.

From Eq. (29), we have

$$\left. \frac{\partial \boldsymbol{f}^{(H)}(\boldsymbol{x}; \boldsymbol{\theta}^{(H)})}{\partial \boldsymbol{a}} \right|_{\boldsymbol{\theta}^{(H)}=\boldsymbol{\theta}_\lambda^{(H)}} = \sum_{j=H_0}^{H} \lambda_j \left. \frac{\partial \boldsymbol{f}^{(H)}(\boldsymbol{x}; \boldsymbol{\theta}^{(H)})}{\partial \boldsymbol{v}_j} \right|_{\boldsymbol{\theta}^{(H)}=\boldsymbol{\theta}_\lambda^{(H)}}$$

$$= \sum_{j=H_0}^{H} \lambda_j \varphi(\boldsymbol{x}; \boldsymbol{w}_j) I \Big|_{\boldsymbol{\theta}^{(H)}=\boldsymbol{\theta}_\lambda^{(H)}}$$

$$= \varphi(\boldsymbol{x}; \boldsymbol{u}_{H_0,*}) I$$

$$= \frac{\partial \boldsymbol{f}^{(H_0)}(\boldsymbol{x}; \boldsymbol{\theta}_*^{(H_0)})}{\partial \boldsymbol{\zeta}_{H_0}},$$

and

$$\left. \frac{\partial \boldsymbol{f}^{(H)}(\boldsymbol{x}; \boldsymbol{\theta}^{(H)})}{\partial \boldsymbol{\xi}_c} \right|_{\boldsymbol{\theta}^{(H)}=\boldsymbol{\theta}_\lambda^{(H)}} = \sum_{k=H_0}^{H} \lambda_k \alpha_{ck} \left. \frac{\partial \boldsymbol{f}^{(H)}(\boldsymbol{x}; \boldsymbol{\theta}^{(H)})}{\partial \boldsymbol{v}_k} \right|_{\boldsymbol{\theta}^{(H)}=\boldsymbol{\theta}_\lambda^{(H)}}$$

$$= \sum_{k=H_0}^{H} \lambda_k \alpha_{ck} \varphi(\boldsymbol{x}; \boldsymbol{u}_{H_0}) I = 0.$$

### D.2 Proof of Lemma 4

We use the notation

$$\boldsymbol{z}_\nu = \boldsymbol{f}^{(H)}(\boldsymbol{x}_\nu; \boldsymbol{\theta}^{(H)}).$$

(i) First, we compute the blocks related to the derivative with respect to $\boldsymbol{\eta}$. We have

$$\frac{\partial L_H(\boldsymbol{\theta}^{(H)})}{\partial \boldsymbol{\eta}_c} = \sum_{\nu=1}^{n} \frac{\partial \ell_\nu(\boldsymbol{\theta}^{(H)})}{\partial \boldsymbol{z}_\nu} \frac{\partial \boldsymbol{z}_\nu}{\partial \boldsymbol{\eta}_c} = \sum_{\nu=1}^{n} \sum_{m=1}^{M} \frac{\partial \ell_\nu(\boldsymbol{\theta}^{(H)})}{\partial z_{\nu,m}} \sum_{j=H_0}^{H} \alpha_{cj} v_{jm} \frac{\partial \varphi(\boldsymbol{x}_\nu; \boldsymbol{w}_j)}{\partial \boldsymbol{w}_j}. \tag{30}$$

It follows from Eqs. (29) and (30) that

$$\frac{\partial^2 L_H(\boldsymbol{\theta}^{(H)})}{\partial \boldsymbol{\eta}_c \partial \boldsymbol{a}} = \sum_{k=H_0}^{H} \lambda_k \frac{\partial^2 L_H(\boldsymbol{\theta}^{(H)})}{\partial \boldsymbol{\eta}_c \partial \boldsymbol{v}_k}$$

$$= \sum_{\nu=1}^{n} \sum_{m=1}^{M} \frac{\partial^2 \ell_\nu(\boldsymbol{\theta}^{(H)})}{\partial \boldsymbol{z}_\nu \partial z_{\nu,m}} \sum_{k=H_0}^{H} \lambda_k \varphi(\boldsymbol{x}_\nu; \boldsymbol{w}_k) \sum_{j=H_0}^{H} \alpha_{cj} v_{jm} \frac{\partial \varphi(\boldsymbol{x}_\nu; \boldsymbol{w}_j)}{\partial \boldsymbol{w}_j}$$

$$+ \sum_{\nu=1}^{n} \frac{\partial \ell_\nu(\boldsymbol{\theta}^{(H)})}{\partial \boldsymbol{z}_\nu} \sum_{k=H_0}^{H} \alpha_{ck} \lambda_k \frac{\partial \varphi(\boldsymbol{x}_\nu; \boldsymbol{w}_k)}{\partial \boldsymbol{w}_k}. \tag{31}$$

By inserting $\boldsymbol{\theta}^{(H)} = \boldsymbol{\theta}_\lambda^{(H)}$, the first term is zero since $\boldsymbol{v}_j = \lambda_j \boldsymbol{\zeta}_*$ and $\sum_j \alpha_{cj} \lambda_j = 0$. The second term is also zero from $\sum_k \alpha_{ck} \lambda_k = 0$.

Differentiation of Eq. (30) with $\boldsymbol{b}$ gives

$$
\frac{\partial^2 L_H(\boldsymbol{\theta}^{(H)})}{\partial \boldsymbol{\eta}_c \partial \boldsymbol{b}} = \sum_{\nu=1}^{n} \sum_{m,m'=1}^{M} \frac{\partial^2 \ell_\nu(\boldsymbol{\theta}^{(H)})}{\partial z_{\nu,m} \partial z_{\nu,m'}} \sum_{k=H_0}^{H} v_{km'} \frac{\partial \varphi(\boldsymbol{x}_\nu; \boldsymbol{w}_k)}{\partial \boldsymbol{w}_k} \sum_{j=H_0}^{H} \alpha_{cj} v_{jm} \frac{\partial \varphi(\boldsymbol{x}_\nu; \boldsymbol{w}_j)}{\partial \boldsymbol{w}_j}
$$
$$
+ \delta_{jk} \sum_{\nu=1}^{n} \sum_{m=1}^{M} \frac{\partial \ell_\nu(\boldsymbol{\theta}^{(H)})}{\partial z_{\nu,m}} \sum_{j=H_0}^{H} \alpha_{cj} v_{jm} \sum_{k=H_0}^{H} \frac{\partial^2 \varphi(\boldsymbol{x}_\nu; \boldsymbol{w}_k)}{\partial \boldsymbol{w}_k \partial \boldsymbol{w}_k}. \tag{32}
$$

At $\boldsymbol{\theta}^{(H)} = \boldsymbol{\theta}_\lambda^{(H)}$, both the terms are zero for the same reason as Eq. (31).

Similarly, for $\boldsymbol{s}_i = \boldsymbol{v}_i$ or $\boldsymbol{w}_i$ $(1 \le i \le H_0 - 1)$,

$$
\frac{\partial^2 L_H(\boldsymbol{\theta}^{(H)})}{\partial \boldsymbol{\eta}_c \partial \boldsymbol{s}_i} = \frac{\partial^2 L_H(\boldsymbol{\theta}^{(H)})}{\partial \boldsymbol{\eta}_c \partial \boldsymbol{s}_i}
$$
$$
= \sum_{\nu=1}^{n} \sum_{m=1}^{M} \frac{\partial^2 \ell_\nu(\boldsymbol{\theta}^{(H)})}{\partial \boldsymbol{z}_\nu \partial z_{\nu,m}} \frac{\partial \boldsymbol{z}_\nu}{\partial \boldsymbol{s}_i} \sum_{j=H_0}^{H} \alpha_{cj} v_{jm} \frac{\partial \varphi(\boldsymbol{x}_\nu; \boldsymbol{w}_j)}{\partial \boldsymbol{w}_j},
$$

which is zero at $\boldsymbol{\theta}^{(H)} = \boldsymbol{\theta}_\lambda^{(H)}$ from $\sum_j \alpha_{cj} \lambda_j = 0$.

Next, from Eqs. (29) and (30), we have

$$
\frac{\partial^2 L_H(\boldsymbol{\theta}^{(H)})}{\partial \boldsymbol{\eta}_c \partial \boldsymbol{\xi}_d} = \sum_{\nu=1}^{n} \frac{\partial^2 \ell_\nu(\boldsymbol{\theta}^{(H)})}{\partial \boldsymbol{z}_\nu \partial \boldsymbol{z}_\nu} \sum_{k=H_0}^{H} \alpha_{dk} \lambda_k \varphi(\boldsymbol{x}_\nu; \boldsymbol{w}_k) \sum_{j=H_0}^{H} \alpha_{cj} v_j \frac{\partial \varphi(\boldsymbol{x}_\nu; \boldsymbol{w}_j)}{\partial \boldsymbol{w}_j}
$$
$$
+ \sum_{\nu=1}^{n} \frac{\partial \ell_\nu(\boldsymbol{\theta}^{(H)})}{\partial \boldsymbol{z}_\nu} \sum_{k=H_0}^{H} \alpha_{dk} \alpha_{ck} \lambda_k \frac{\partial \varphi(\boldsymbol{x}_\nu; \boldsymbol{w}_k)}{\partial \boldsymbol{w}_k}. \tag{33}
$$

At $\boldsymbol{\theta}^{(H)} = \boldsymbol{\theta}_\lambda^{(H)}$, the first trem vanishes and the second term reduces to

$$
\frac{\partial^2 L_H(\boldsymbol{\theta}_\lambda^{(H)})}{\partial \boldsymbol{\eta}_c \partial \boldsymbol{\xi}_d} = (A\Lambda A^T)_{cd} \sum_{\nu=1}^{n} \frac{\partial \ell_\nu(\boldsymbol{\theta}_*^{(H_0)})}{\partial \boldsymbol{z}_\nu} \frac{\partial \varphi(\boldsymbol{x}_\nu; \boldsymbol{u}_{H_0,*})}{\partial \boldsymbol{u}_{H_0}},
$$

which is $(A\Lambda A^T)_{cd} F$.

The block $\frac{L_H(\boldsymbol{\theta}_\lambda^{(H)})}{\partial \boldsymbol{\eta}_c \partial \boldsymbol{\eta}_d}$ can be computed in a similar way to Eq. (33):

$$
\frac{\partial^2 L_H(\boldsymbol{\theta}^{(H)})}{\partial \boldsymbol{\eta}_c \partial \boldsymbol{\eta}_d} = \sum_{\nu=1}^{n} \sum_{m,m'=1}^{M} \frac{\partial^2 \ell_\nu(\boldsymbol{\theta}^{(H)})}{\partial z_{\nu,m'} \partial z_{\nu,m}} \sum_{j=H_0}^{H} \alpha_{cj} v_{jm} \frac{\partial \varphi(\boldsymbol{x}_\nu; \boldsymbol{w}_j)}{\partial \boldsymbol{w}_j} \sum_{k=H_0}^{H} \alpha_{dk} v_{km'} \frac{\partial \varphi(\boldsymbol{x}_\nu; \boldsymbol{w}_k)}{\partial \boldsymbol{w}_k}
$$
$$
+ \sum_{\nu=1}^{n} \sum_{m=1}^{M} \frac{\partial \ell_\nu(\boldsymbol{\theta}^{(H)})}{\partial z_{\nu,m}} \sum_{j=H_0}^{H} \alpha_{cj} \alpha_{dj} v_{jm} \frac{\partial^2 \varphi(\boldsymbol{x}_\nu; \boldsymbol{w}_j)}{\partial \boldsymbol{w}_j \boldsymbol{w}_j}.
$$

By plugging $\boldsymbol{\theta}^{(H)} = \boldsymbol{\theta}_\lambda^{(H)}$, the first term is zero, and the second term is reduced to

$$
\sum_{j=H_0}^{H} \lambda_j \alpha_{cj} \alpha_{dj} \sum_{\nu=1}^{n} \frac{\partial \ell_\nu(\boldsymbol{\theta}^{(H)})}{\partial \boldsymbol{z}_\nu} \boldsymbol{\zeta}_{H_0,*} \frac{\partial^2 \varphi(\boldsymbol{x}_\nu; \boldsymbol{u}_{H_0,*})}{\partial \boldsymbol{u}_{H_0} \partial \boldsymbol{u}_{H_0}}, \tag{34}
$$

which is $(A\Lambda A^T)_{cd} G$.

(ii) Second, we will compute the remaining second derivatives including $\boldsymbol{\xi}_c$. From Eq. (29), the first derivative with respect to $\boldsymbol{\xi}_c$ is given by

$$
\frac{\partial L_H(\boldsymbol{\theta}^{(H)})}{\partial \boldsymbol{\xi}_c} = \sum_{\nu=1}^{n} \frac{\partial \ell_\nu(\boldsymbol{\theta}^{(H)})}{\partial \boldsymbol{z}_\nu} \sum_{j=H_0}^{H} \lambda_j \alpha_{cj} \varphi(\boldsymbol{x}_\nu; \boldsymbol{w}_j). \tag{35}
$$

From this expression,

$$\frac{\partial^2 L_H(\boldsymbol{\theta}^{(H)})}{\partial \boldsymbol{\xi}_c \partial \boldsymbol{v}_k}\bigg|_{\boldsymbol{\theta}^{(H)}=\boldsymbol{\theta}_\lambda^{(H)}} = \sum_{\nu=1}^n \frac{\partial^2 \ell_\nu(\boldsymbol{\theta}_\lambda^{(H)})}{\partial \boldsymbol{z}_\nu \partial \boldsymbol{z}_\nu} \sum_{j=H_0}^H \lambda_j \alpha_{cj} \big(\varphi(\boldsymbol{x}_\nu; \boldsymbol{u}_{H_0,*})\big)^2$$

$$= 0,$$

which means $\frac{\partial^2 L_H(\boldsymbol{\theta}_\lambda^{(H)})}{\partial \boldsymbol{\xi}_c \partial \boldsymbol{\xi}_d}$ and $\frac{\partial^2 L_H(\boldsymbol{\theta}_\lambda^{(H)})}{\partial \boldsymbol{\xi}_c \partial \boldsymbol{a}}$ are zero.

It follows from Eqs. (35) and (29) that

$$\frac{\partial^2 L_H(\boldsymbol{\theta}^{(H)})}{\partial \boldsymbol{\xi}_c \partial \boldsymbol{b}}\bigg|_{\boldsymbol{\theta}^{(H)}=\boldsymbol{\theta}_\lambda^{(H)}}$$

$$= \sum_{\nu=1}^n \sum_{m=1}^M \frac{\partial^2 \ell_\nu(\boldsymbol{\theta}_\lambda^{(H)})}{\partial \boldsymbol{z}_\nu \partial \boldsymbol{z}_{\nu,m}} \sum_{j=H_0}^H \lambda_j \alpha_{cj} \varphi(\boldsymbol{x}_\nu; \boldsymbol{u}_{H_0,*}) \sum_{k=H_0}^H v_{km} \frac{\partial \varphi(\boldsymbol{x}_\nu; \boldsymbol{u}_{H_0,*})}{\partial \boldsymbol{u}_{H_0}}$$

$$+ \sum_{\nu=1}^n \frac{\partial \ell_\nu(\boldsymbol{\theta}^{(H)})}{\partial \boldsymbol{z}_\nu} \sum_{j=H_0}^H \lambda_j \alpha_{cj} \frac{\partial \varphi(\boldsymbol{x}_\nu; \boldsymbol{u}_{H_0,*})}{\partial \boldsymbol{u}_{H_0}},$$

which is zero from $\sum_j \alpha_{cj} \lambda_j = 0$.

It is also easy to see that for $\boldsymbol{s}_i = \boldsymbol{v}_i$ or $\boldsymbol{w}_i$ ($1 \le i \le H_0 - 1$)

$$\frac{\partial^2 L_H(\boldsymbol{\theta}^{(H)})}{\partial \boldsymbol{\xi}_c \partial \boldsymbol{s}_i}\bigg|_{\boldsymbol{\theta}^{(H)}=\boldsymbol{\theta}_\lambda^{(H)}} = \boldsymbol{0}.$$

(III) We compute the upper-left four blocks. We have

$$\frac{\partial L_H(\boldsymbol{\theta}^{(H)})}{\partial \boldsymbol{a}} = \sum_{\nu=1}^n \frac{\partial \ell_\nu(\boldsymbol{\theta}^{(H)})}{\partial \boldsymbol{z}_\nu} \sum_{j=H_0}^H \lambda_j \varphi(\boldsymbol{x}_\nu; \boldsymbol{w}_j), \qquad (36)$$

from which

$$\frac{\partial^2 L_H(\boldsymbol{\theta}^{(H)})}{\partial \boldsymbol{a} \partial \boldsymbol{a}}\bigg|_{\boldsymbol{\theta}^{(H)}=\boldsymbol{\theta}_\lambda^{(H)}} = \sum_{\nu=1}^n \frac{\partial^2 \ell_\nu(\boldsymbol{\theta}_*^{(H_0)})}{\partial \boldsymbol{z}_\nu \partial \boldsymbol{z}_\nu} \varphi(\boldsymbol{x}_\nu; \boldsymbol{u}_{H_0,*})^2 = \frac{\partial^2 L_{H_0}(\boldsymbol{\theta}_*^{(H_0)})}{\partial \boldsymbol{\zeta}_{H_0} \partial \boldsymbol{\zeta}_{H_0}}$$

and

$$\frac{\partial^2 L_H(\boldsymbol{\theta}^{(H)})}{\partial \boldsymbol{a} \partial \boldsymbol{b}}\bigg|_{\boldsymbol{\theta}^{(H)}=\boldsymbol{\theta}_\lambda^{(H)}}$$

$$= \sum_{\nu=1}^n \sum_{m=1}^M \frac{\partial^2 \ell_\nu(\boldsymbol{\theta}^{(H)})}{\partial \boldsymbol{z}_\nu \partial \boldsymbol{z}_{\nu,m}} \sum_{j=H_0}^H \lambda_j \varphi(\boldsymbol{x}_\nu; \boldsymbol{w}_j) \sum_{k=H_0}^H v_{km} \frac{\partial \varphi(\boldsymbol{x}_\nu; \boldsymbol{w}_k)}{\partial \boldsymbol{w}_k}\bigg|_{\boldsymbol{\theta}^{(H)}=\boldsymbol{\theta}_\lambda^{(H)}}$$

$$+ \sum_{\nu=1}^n \frac{\partial \ell_\nu(\boldsymbol{\theta}^{(H)})}{\partial \boldsymbol{z}_\nu} \sum_{j=H_0}^H \lambda_j \frac{\partial \varphi(\boldsymbol{x}_\nu; \boldsymbol{w}_j)}{\partial \boldsymbol{w}_j}\bigg|_{\boldsymbol{\theta}^{(H)}=\boldsymbol{\theta}_\lambda^{(H)}}$$

$$= \sum_{\nu=1}^n \frac{\partial^2 \ell_\nu(\boldsymbol{\theta}_*^{(H_0)})}{\partial \boldsymbol{z}_\nu \partial \boldsymbol{z}_\nu} \boldsymbol{\zeta}_{H_0,*} \varphi(\boldsymbol{x}_\nu; \boldsymbol{u}_{H_0,*}) \frac{\partial \varphi(\boldsymbol{x}_\nu; \boldsymbol{u}_{H_0,*})}{\partial \boldsymbol{u}_{H_0}} + \sum_{\nu=1}^n \frac{\partial \ell_\nu(\boldsymbol{\theta}_*^{(H_0)})}{\partial \boldsymbol{z}_\nu} \frac{\partial \varphi(\boldsymbol{x}_\nu; \boldsymbol{u}_{H_0,*})}{\partial \boldsymbol{u}_{H_0}}$$

$$= \frac{\partial^2 L_{H_0}(\boldsymbol{\theta}_*^{(H_0)})}{\partial \boldsymbol{\zeta}_{H_0} \partial \boldsymbol{u}_{H_0}}.$$

Finally, using

$$\frac{\partial L_H(\boldsymbol{\theta}^{(H)})}{\partial \boldsymbol{b}} = \sum_{\nu=1}^n \frac{\partial \ell_\nu(\boldsymbol{\theta}^{(H)})}{\partial \boldsymbol{z}_\nu} \frac{\partial \boldsymbol{z}_\nu}{\partial \boldsymbol{b}} = \sum_{\nu=1}^n \sum_{m=1}^M \frac{\partial \ell_\nu(\boldsymbol{\theta}^{(H)})}{\partial \boldsymbol{z}_{\nu,m}} \sum_{j=H_0}^H v_{jm} \frac{\partial \varphi(\boldsymbol{x}_\nu; \boldsymbol{w}_j)}{\partial \boldsymbol{w}_j},$$

we have

$$
\frac{\partial^2 L_H(\boldsymbol{\theta}^{(H)})}{\partial \boldsymbol{b} \partial \boldsymbol{b}}\Bigg|_{\boldsymbol{\theta}^{(H)}=\boldsymbol{\theta}_\lambda^{(H)}}
$$

$$
= \sum_{\nu=1}^{n} \sum_{m,m'=1}^{M} \frac{\partial^2 \ell_\nu(\boldsymbol{\theta}^{(H)})}{\partial z_{\nu,m} \partial z_{\nu,m'}} \sum_{j=H_0}^{H} v_{jm} \frac{\partial \varphi(\boldsymbol{x}_\nu; \boldsymbol{w}_j)}{\partial \boldsymbol{w}_j} \sum_{k=1}^{H} v_{km'} \frac{\partial \varphi(\boldsymbol{x}_\nu; \boldsymbol{w}_k)}{\partial \boldsymbol{w}_k}\Bigg|_{\boldsymbol{\theta}^{(H)}=\boldsymbol{\theta}_\lambda^{(H)}}
$$

$$
+ \sum_{\nu=1}^{n} \sum_{m=1}^{M} \frac{\partial \ell_\nu(\boldsymbol{\theta}^{(H)})}{\partial z_{\nu,m}} \sum_{j=H_0}^{H} v_{jm} \frac{\partial^2 \varphi(\boldsymbol{x}_\nu; \boldsymbol{w}_j)}{\partial \boldsymbol{w}_j \partial \boldsymbol{w}_j}\Bigg|_{\boldsymbol{\theta}^{(H)}=\boldsymbol{\theta}_\lambda^{(H)}}
$$

$$
= \sum_{\nu=1}^{n} \left( \boldsymbol{\zeta}_{H_0,*}^{T} \frac{\partial^2 \ell_\nu(\boldsymbol{\theta}_*^{(H_0)})}{\partial \boldsymbol{z} \partial \boldsymbol{z}} \boldsymbol{\zeta}_{H_0,*} \right) \frac{\partial \varphi(\boldsymbol{x}_\nu; \boldsymbol{u}_{H_0,*})}{\partial \boldsymbol{u}_{H_0}} \frac{\partial \varphi(\boldsymbol{x}_\nu; \boldsymbol{u}_{H_0,*})}{\partial \boldsymbol{u}_{H_0}}
$$

$$
+ \sum_{\nu=1}^{n} \frac{\partial \ell_\nu(\boldsymbol{\theta}^{(H)})}{\partial \boldsymbol{z}_\nu} \boldsymbol{\zeta}_{H_0,*} \frac{\partial^2 \varphi(\boldsymbol{x}_\nu; \boldsymbol{u}_{H_0,*})}{\partial \boldsymbol{u}_{H_0} \boldsymbol{u}_{H_0}}
$$

$$
= \frac{\partial^2 L_{H_0}(\boldsymbol{\theta}_*^{(H_0)})}{\partial \boldsymbol{u}_{H_0} \partial \boldsymbol{u}_{H_0}}.
$$

(iv) Finally, it is similarly proved that for $\boldsymbol{s}_i = \boldsymbol{v}_i$ or $\boldsymbol{w}_i$ ($1 \le i \le H_0 - 1$)

$$
\frac{\partial^2 L_H(\boldsymbol{\theta}^{(H)})}{\partial \boldsymbol{a} \partial \boldsymbol{s}_i}\Bigg|_{\boldsymbol{\theta}^{(H)}=\boldsymbol{\theta}_\lambda^{(H)}} = \frac{\partial^2 L_{H_0}(\boldsymbol{\theta}_*^{(H_0)})}{\partial \boldsymbol{\zeta}_{H_0} \partial \boldsymbol{s}_i}
$$

$$
\frac{\partial^2 L_H(\boldsymbol{\theta}^{(H)})}{\partial \boldsymbol{b} \partial \boldsymbol{s}_i}\Bigg|_{\boldsymbol{\theta}^{(H)}=\boldsymbol{\theta}_\lambda^{(H)}} = \frac{\partial^2 L_{H_0}(\boldsymbol{\theta}_*^{(H_0)})}{\partial \boldsymbol{u}_{H_0} \partial \boldsymbol{s}_i}.
$$

This completes the proof.

### D.3 Proof of Theorem 5

Let $\tilde{F} := (A\Lambda A^T) \otimes F$ and $\tilde{G} := (A\Lambda A^T) \otimes G$. Since $\lambda_j \ne 0$ ($\forall j$) and $A$ is of full rank, $(A\Lambda A^T)$ is of full rank. (i) Under the assumption, $\tilde{G}$ is invertible. Then, the lower-right four blocks of the Hessian has the expression

$$
\begin{pmatrix} I & -\tilde{F}\tilde{G}^{-1} \\ O & I \end{pmatrix} \begin{pmatrix} O & \tilde{F} \\ \tilde{F}^T & \tilde{G} \end{pmatrix} \begin{pmatrix} I & O \\ -\tilde{F}\tilde{G}^{-1} & I \end{pmatrix} = \begin{pmatrix} -\tilde{F}^T \tilde{G}^{-1} \tilde{F} & O \\ O & \tilde{G}. \end{pmatrix}. \tag{37}
$$

If $G$ is positive definite, so is $\tilde{G}$, and thus $-\tilde{F}^T \tilde{G}^{-1} \tilde{F}$ has negative eigenvalues for $F \ne O$. The Hessian of $L_H$ at $\boldsymbol{\theta}_\lambda^{(H)}$ has both of positive and negative eigenvalues, which implies $\boldsymbol{\theta}_\lambda^{(H)}$ is a saddle point. The case of negative definite $G$ is similar. (ii) If $G$ has positive and negative definite, so does $\tilde{G}$. This means that the Hessian of $L_H$ at $\boldsymbol{\theta}_\lambda^{(H)}$ has positive and negative eigenvalues. □

*Remark.* If $F$ is of full rank, which is $r = (H - H_0) \min\{D, M\}$, then the matrix $\tilde{F}^T \tilde{G}^{-1} \tilde{F}$ has $r$ positive eigenvalues. Thus, the number of positive and negative eigenvalues of the matrix in Eq. (37) are $(H - H_0) \times D$ and $r$, respectively. When $F$ has positive and negative engenvalues, the index depends on the eigenvectors of $\tilde{F}$ and $\tilde{G}$, and not easy to tell.

## E  Local minima for smooth networks of 1-dimensional output

The special property of $M = 1$ is caused by vanishing $\tilde{F}$ in the Hessian. In fact, the stationarity condition $\frac{\partial L_{H_0}(\boldsymbol{\theta}_*^{(H_0)})}{\partial \boldsymbol{u}_{H_0}} = 0$ implies

$$
\boldsymbol{\zeta}_{H_0,*} \sum_{\nu=1}^{n} \frac{\partial \ell_\nu(\boldsymbol{\theta}_*^{(H_0)})}{\partial z_\nu} \frac{\partial \varphi(\boldsymbol{x}_\nu; \boldsymbol{u}_{H_0*})}{\partial \boldsymbol{u}_{H_0}} = 0.
$$

Note that $\zeta_{H_0}$ is a scalar, and if we assume $\zeta_{H_0*} \neq 0$, the above condition implies $F = 0$. Then the corresponding part of the Hessian takes the form

$$\begin{pmatrix} O & O \\ O & \tilde{G} \end{pmatrix},$$

which does not have negative eigenvalues if $G$ is non-negative definite. The zero blocks of the Hessian correspond to the directions $\boldsymbol{\xi}_c$ ($c = H_0 + 1, \ldots, H$), which make an affine subspace of $\Pi_{repl}(\boldsymbol{\theta}_{\boldsymbol{\lambda}}^{(H_0)})$ having the same value $L_H(\boldsymbol{\theta}^{(H)}) = L_{H_0}(\boldsymbol{\theta}_*^{(H_0)})$. Therefore, only the Hessian in the directions $(\boldsymbol{a}, \boldsymbol{b}, \boldsymbol{\eta}_{H_0+1}, \ldots, \boldsymbol{\eta}_H)$ matters to determine if $\boldsymbol{\theta}_{\boldsymbol{\lambda}}^{(H)}$ is a minimum or saddle point. Note also that for $M \geq 2$ the stationarity condition gives

$$\sum_{\nu=1}^{n} \sum_{m=1}^{M} \frac{\partial \ell_\nu(\boldsymbol{\theta}_*^{(H_0)})}{\partial z_{\nu,m}} \zeta_{H_0,m*} \frac{\partial \varphi(\boldsymbol{x}_\nu; \boldsymbol{u}_{H_0,*})}{\partial \boldsymbol{u}_{H_0}} = 0,$$

which does not necessary mean $F = O$.

The following theorem is a slight extension of Fukumizu and Amari [2, Theorem 3], in which only the case $H = H_0 + 1$ is discussed.

**Theorem 11.** *Suppose that the dimension of the output is 1 and $\boldsymbol{\theta}_*^{(H_0)}$ is a minimum of $L_{H_0}$ with positive definite Hessian matrix. In the following, the matrix $G$ and the parameter $\boldsymbol{\theta}_{\boldsymbol{\lambda}}^{(H)}$ are used in the same meaning as in Lemma 4.*

*(1) Assume that the matrix $G$ is positive definite.*

    *(a) $\boldsymbol{\theta}_{\boldsymbol{\lambda}}^{(H)}$ with $\sum_{j=H_0}^{H} \lambda_j = 1$ and $\lambda_j > 0$ ($\forall j$) is a minimum of $L_H$.*

    *(b) $\boldsymbol{\theta}_{\boldsymbol{\lambda}}^{(H)}$ with $\sum_{j=H_0}^{H} \lambda_j = 1$ and $\lambda_j < 0$ for some $j$ is a saddle point of $L_H$.*

*(2) Assume that the matrix $G$ is negative definite.*

    *(a) If $\sum_{j=H_0}^{H} \lambda_j = 1$ and there is only one $i_0$ such that $\lambda_{i_0} > 0$ and $\lambda_j < 0$ ($\forall j \neq i_0$), $\boldsymbol{\theta}_{\boldsymbol{\lambda}}^{(H)}$ is a minimum of $L_H$.*

    *(b) If $\sum_{j=H_0}^{H} \lambda_j = 1$ and $\lambda_j > 0$ for at least two indices, $\boldsymbol{\theta}_{\boldsymbol{\lambda}}^{(H)}$ is a saddle point of $L_H$.*

*(3) If the matrix $G$ has both of positive and negative eigenvalues, $\boldsymbol{\theta}_{\boldsymbol{\lambda}}^{(H)}$ is a saddle point for any $\boldsymbol{\lambda}$ with $\sum_{a=H_0}^{H} \lambda_a = 1$ and $\lambda_a \neq 0$ ($\forall a$).*

*Proof.* For notational simplicity, the proof is given only for $H_0 = 1$; $\zeta_1$ and $\boldsymbol{u}_1$ are written by $\zeta$ and $\boldsymbol{u}$, respectively. Extension to a general $H_0$ is easy and we omit it. In the proof, let $\tilde{A}^T := (\boldsymbol{1}_H A^T)$, which is invertible by assumption. Note also that $\zeta, v_j$ are scalar parameters in the case of $M = 1$.

(1-a). We first show that if $G$ is positive definite, the lower-right block of the Hessian, $\frac{\partial^2 L_H(\boldsymbol{\theta}_{\boldsymbol{\lambda}}^{(H)})}{\partial \boldsymbol{\eta} \partial \boldsymbol{\eta}} = (A\Lambda A^T) \otimes G$, is positive definite. This can be proved if $A\Lambda A^T$ is positive definite, since the eigenvalues of the tensor product is given by the products of respective eigenvalues of $A\Lambda A^T$ and $G$. By the assumptions, $A\Lambda A^T$ is non-negative definite. Suppose $A\Lambda A^T \boldsymbol{s} = 0$ for $\boldsymbol{s} \in \mathbb{R}^{H-1} \backslash \{0\}$. Then, $A^T \boldsymbol{s} = 0$, and this implies $\tilde{A}^T \tilde{\boldsymbol{s}} = 0$ for $\tilde{\boldsymbol{s}} = (\boldsymbol{s}^T, 0)^T \in \mathbb{R}^H$. This is impossible by the invertible assumption of $\tilde{A}$.

Now consider the Hessian $\nabla^2 L_H(\boldsymbol{\theta}_{\boldsymbol{\lambda}}^{(H)})$ in Lemma 4. It is obvious that this Hessian is non-negative definite, but not positive definite, as the blocks corresponding to $(\xi_j)_{j=2}^{H}$ are zero. Let $\Pi_{\boldsymbol{\theta}_*^{(H_0)}}$ be the $(H-1)$ dimensional affine plane in the parameter space of $\mathcal{N}_H$ such that

$$\Pi_{\boldsymbol{\theta}_*^{(H_0)}} := \{(a, \xi_2, \ldots, \xi_H; \boldsymbol{b}, \boldsymbol{\eta}_2, \ldots, \boldsymbol{\eta}_H) \mid a = \zeta_*, \boldsymbol{b} = \boldsymbol{u}_*, \boldsymbol{\eta}_2 = \cdots = \boldsymbol{\eta}_H = 0\}.$$

This plane includes $\boldsymbol{\theta}_{\boldsymbol{\lambda}}^{(H)}$, and is parallel to the subspace spanned by $\xi_j$ axes. The function $L_H$ takes the same value as $L_1(\boldsymbol{\theta}_*^{(1)})$ on the whole of $\Pi_{\boldsymbol{\theta}_*^{(H_0)}}$. Thus, $\boldsymbol{\theta}_{\boldsymbol{\lambda}}^{(H)}$ is a minimum of $L_H$ if the Hessian

$$\Pi := \big\{ \boldsymbol{\theta}^{(H)} \mid \boldsymbol{a} = \boldsymbol{\zeta}, \; \boldsymbol{b} = \boldsymbol{u}, \; \boldsymbol{\eta}_c = 0 \big\}$$

Figure 4: All the parameters on the affine subspace $\Pi$ has the same function as $\boldsymbol{f}^{(H)}(\boldsymbol{x}; \boldsymbol{\theta}_\lambda^{(H)})$, and the affine subspace (in red) is a set of stationary points of $L_H(\boldsymbol{\theta}^{(H)})$. The local behavior of $L_H$ around $\boldsymbol{\theta}_\lambda^{(H)}$ is determined by the second derivative along the $\boldsymbol{a}, \boldsymbol{b}, \boldsymbol{\eta}_c$ directions.

is positive definite along the directions compliment to $\Pi_{\boldsymbol{\theta}_*^{(H_0)}}$ (see Figure 4). From Lemma 4, the Hessian at $\boldsymbol{\theta}_\lambda^{(H)}$ along the directions $(\boldsymbol{a}, \boldsymbol{b}, \boldsymbol{\eta}_j)$ is given by

$$\begin{pmatrix} \frac{\partial^2 L_1(\boldsymbol{\theta}_*^{(1)})}{\partial \boldsymbol{\theta}_*^{(H_0)} \partial \boldsymbol{\theta}_*^{(H_0)}} & O \\ O & (A \Lambda A^T) \otimes G \end{pmatrix},$$

which is positive definite. This completes the proof of (1-a).

(1-b) From $A\lambda = 0$, it is easy to see that

$$\tilde{A} \Lambda \tilde{A}^T = \begin{pmatrix} 1 & 0 \\ 0^T & A\Lambda A^T \end{pmatrix}.$$

Thus, the eigenvalues of $\tilde{A}\Lambda\tilde{A}^T$ is the eigenvalues of $A\Lambda A^T$ and 1. By Sylvester's law of inertia, the signature (the pair of the number of positive eigenvalues and that of negative ones) of $\tilde{A}\Lambda\tilde{A}$ coincides with the signature of $\Lambda$. Since some $\lambda_i$ are negative by the assumption, $A\Lambda A^T$ has a negative eigenvalue. Thus, under the assumption that $G$ is positive definite, $(A\Lambda A^T) \otimes G$ has a negative eigenvalue. Since $\frac{\partial^2 L_H(\boldsymbol{\theta}_\lambda^{(H)})}{\partial \boldsymbol{a} \partial \boldsymbol{a}}$ is positive definite, the Hessian of $L_H(\boldsymbol{\theta}^{(H)})$ at $\boldsymbol{\theta}^{(H)} = \boldsymbol{\theta}_\lambda^{(H)}$ has positive and negative eigenvalues, which means $\boldsymbol{\theta}_\lambda^{(H)}$ is a saddle point.

(2-a) It suffices to show that $A\Lambda A^T$ is negative definite. Then, $(A\Lambda A^T) \otimes G$ is positive definite, and the assertion is proved by the same argument as (1-a). Without loss of generality, we can assume that $\lambda_j < 0$ for $1 \le j \le H-1$ and $\lambda_H > 0$. Let $A = (A_0, \boldsymbol{h})$ where $A_0$ is an invertible matrix of size $H-1$, and let $\boldsymbol{\lambda}^T = (\boldsymbol{\lambda}_0^T, \lambda_H)$ with $\boldsymbol{\lambda}_0 \in \mathbb{R}^{H-1}$. The elements of $\boldsymbol{\lambda}_0$ are all negative by assumption. It follows that

$$A_0 \boldsymbol{\lambda}_0 + \lambda_H \boldsymbol{h} = \boldsymbol{0}, \qquad \sum_{j=1}^{H} \lambda_j = 1.$$

A simple computation using $\boldsymbol{h} = -\frac{1}{\lambda_H} A_0 \boldsymbol{\lambda}_0$ provides

$$A \Lambda A^T = A_0 \Big( \Lambda_0 + \frac{1}{\lambda_H} \boldsymbol{\lambda}_0 \boldsymbol{\lambda}_0^T \Big) A_0^T,$$

where $\Lambda_0 = \mathrm{Diag}(\lambda_1, \ldots, \lambda_{H-1})$. It is then sufficient to show that $B_0 := \Lambda_0 + \frac{1}{\lambda_H} \boldsymbol{\lambda}_0 \boldsymbol{\lambda}_0^T$ is negative definite. If $\boldsymbol{s} \in \mathbb{R}^{H-1} \backslash \{0\}$ is orthogonal to $\boldsymbol{\lambda}_0$, we have $\boldsymbol{s}^T B_0 \boldsymbol{s} = \boldsymbol{s}^T \Lambda_0 \boldsymbol{s} < 0$. Additionally,

$$
\begin{aligned}
\boldsymbol{\lambda}_0^T B_0 \boldsymbol{\lambda}_0 &= \sum_{j=1}^{H-1} \lambda_j^3 + \frac{1}{\lambda_H} \Big( \sum_{j=1}^{H-1} \lambda_j^2 \Big)^2 \\
&= \frac{1}{\lambda_H} \Big\{ \Big( 1 - \sum_{j=1}^{H-1} \lambda_j \Big) \Big( \sum_{j=1}^{H-1} \lambda_j^3 \Big) + \Big( \sum_{j=1}^{H-1} \lambda_j^2 \Big)^2 \Big\} \\
&= \frac{1}{\lambda_H} \Big\{ \Big( \sum_{j=1}^{H-1} \lambda_j^3 \Big) + \sum_{i \neq j} \lambda_i^2 \lambda_j^2 - \sum_{i \neq j} \lambda_i \lambda_j^3 \Big\} \\
&= \frac{1}{\lambda_H} \Big\{ \Big( \sum_{j=1}^{H-1} \lambda_j^3 \Big) + \sum_{i \neq j} \lambda_i^2 \lambda_j^2 - \sum_{i \neq j} \lambda_i \lambda_j \frac{\lambda_i^2 + \lambda_j^2}{2} \Big\} \\
&= \frac{1}{\lambda_H} \Big\{ \Big( \sum_{j=1}^{H-1} \lambda_j^3 \Big) - \sum_{i \neq j} \frac{1}{2} \lambda_i \lambda_j (\lambda_i - \lambda_j)^2 \Big\},
\end{aligned}
$$

which is negative as well. This proves the assertion.

(2-b) If there are two positive eigenvalues, the corresponding eigenspaces of at least two dimensions must intersects with the $H - 1$ dimensional subspace spanned by the row vectors of $A$. Thus, $A \Lambda A^T$ has at least one positive eigenvalue, which means $(A \Lambda A^T) \otimes G$ has negative eigenvalues. The remaining proof is similar to (1-b).

(3) $A \Lambda A^T$ is of full rank, and thus $(A \Lambda A^T) \otimes G$ has both of positive and negative eigenvalues. The assertion is proved by the same argument as the case (1-b). $\qquad\square$

## F   Proof of Proposition 8 and Theorem 10 in Section 4

### F.1   Proof of Proposition 8

First, note that, from $\boldsymbol{u}_{H_0 *}^T \boldsymbol{x}_\nu \neq 0 (\forall \nu)$, there is $\delta > 0$ such that for each $\boldsymbol{x}_\nu$ the sign of $(\boldsymbol{u}_{H_0, *} + \sum_{c=H_0+1}^{H} \alpha_{cj} \boldsymbol{\eta}_c)^T \boldsymbol{x}_\nu$ equals to that of $\boldsymbol{u}_{H_0, *}^T \boldsymbol{x}_\nu$ for any $j = H_0, \ldots, H$ and $(\boldsymbol{\eta}_c)_{c=H_0+1}^{H}$ such that $\|(\boldsymbol{\eta}_{H_0+1}, \cdots, \boldsymbol{\eta}_H)\| \leq \delta$.

Fix $\boldsymbol{x}_\nu$, and assume first $\boldsymbol{u}_{H_0, *}^T \boldsymbol{x}_\nu > 0$. Then, $(\boldsymbol{u}_{H_0, *} + \sum_{c=H_0+1}^{H} \alpha_{cj} \boldsymbol{\eta}_c)^T \boldsymbol{x}_\nu > 0$ holds for $(\boldsymbol{\eta}_c)_c$ with $\|(\boldsymbol{\eta}_c)_c\| \leq \delta$. With the notation

$$
\mathcal{F}_{H_0} := \sum_{i=1}^{H_0-1} \boldsymbol{v}_i \varphi(\boldsymbol{x}_\nu; \boldsymbol{w}_i) = \sum_{i=1}^{H_0-1} \boldsymbol{\zeta}_{i, *} \varphi(\boldsymbol{x}_\nu; \boldsymbol{u}_{i, *}), \tag{38}
$$

for any $\boldsymbol{\theta}^{(H)} \in B_\delta^{\boldsymbol{\eta}}(\boldsymbol{\theta}_{\gamma, \boldsymbol{\beta}}^{(H)})$, we have

$$
\begin{aligned}
\boldsymbol{f}^{(H)}(\boldsymbol{x}_\nu; \boldsymbol{\theta}^{(H)}) &= \mathcal{F}_{H_0} + \sum_{j=H_0}^{H} \gamma_j \boldsymbol{\zeta}_{H_0, *} \varphi \Big( \beta_j \big( \boldsymbol{u}_{H_0, *} + \sum_{c=H_0+1}^{H} \alpha_{cj} \boldsymbol{\eta}_c \big)^T \boldsymbol{x}_\nu \Big) \\
&= \mathcal{F}_{H_0} + \sum_{j=H_0}^{H} \gamma_j \boldsymbol{\zeta}_{H_0, *} \beta_j \big( \boldsymbol{u}_{H_0, *} + \sum_{c=H_0+1}^{H} \alpha_{cj} \boldsymbol{\eta}_c \big)^T \boldsymbol{x}_\nu \\
&= \mathcal{F}_{H_0} + \sum_{j=H_0}^{H} \gamma_j \beta_j \boldsymbol{\zeta}_{H_0, *} \boldsymbol{u}_{H_0, *}^T \boldsymbol{x}_\nu + \boldsymbol{\zeta}_{H_0, *} \sum_{c=H_0+1}^{H} \sum_{j=H_0}^{H} \alpha_{cj} \gamma_j \beta_j \boldsymbol{\eta}_c^T \boldsymbol{x}_\nu \\
&= \mathcal{F}_{H_0} + \boldsymbol{\zeta}_{H_0, *} \boldsymbol{u}_{H_0, *}^T \boldsymbol{x}_\nu \\
&= \boldsymbol{f}^{(H_0)}(\boldsymbol{x}_\nu; \boldsymbol{\theta}_*^{(H_0)}),
\end{aligned}
$$

where we used $\sum_j \gamma_j \beta_j = 1$ and $\sum_j \alpha_{cj} \gamma_j \beta_j = 0$.

Next, if $\boldsymbol{u}_{H_0,*}^T \boldsymbol{x}_\nu < 0$, we have

$$\boldsymbol{f}^{(H_0)}(\boldsymbol{x}_\nu; \boldsymbol{\theta}_*^{(H_0)}) = \mathcal{F}_{H_0},$$

and

$$\boldsymbol{f}^{(H)}(\boldsymbol{x}_\nu; \boldsymbol{\theta}^{(H)}) = \mathcal{F}_{H_0} + \sum_{j=H_0}^{H} \gamma_j \boldsymbol{\zeta}_{H_0,*} \, \varphi\Big(\beta_j \big(\boldsymbol{u}_{H_0,*} + \sum_{c=H_0+1}^{H} \alpha_{cj} \boldsymbol{\eta}_c\big)^T \boldsymbol{x}_\nu\Big) = \mathcal{F}_{H_0},$$

which completes the proof.

### F.2 Proof of Theorem 10

We use the same reparameterization $(\boldsymbol{v}_1, \ldots, \boldsymbol{v}_{H_0-1}, \boldsymbol{a}, \boldsymbol{w}_1, \ldots, \boldsymbol{w}_{H_0-1}, \boldsymbol{b}, \boldsymbol{\xi}_{H_0+1}, \ldots, \boldsymbol{\xi}_H, \boldsymbol{\eta}_{H_0+1}, \ldots, \boldsymbol{\eta}_H)$ as in Section 4.1 with $A\gamma = 0$. We focus on the behavior of $L_H$ for a change of $\boldsymbol{\xi}_c, \boldsymbol{\eta}_c$ with the others fixed at the values of $\boldsymbol{\theta}_\gamma^{(H)}$. Note that, by the assumption $\boldsymbol{u}_{H_0}^T \boldsymbol{x}_\nu \neq 0$ for any $\nu$, $L_H(\boldsymbol{\theta}^{(H)})$ is differentiable at $\boldsymbol{\theta}_\gamma^{(H)}$ with respect to $\boldsymbol{\xi}_c, \boldsymbol{\eta}_c$. By the same manner as Lemma 3, we have

$$\frac{\partial L_H(\boldsymbol{\theta}^{(H)})}{\partial \boldsymbol{\eta}_c}\Big|_{\boldsymbol{\theta}^{(H)}=\boldsymbol{\theta}_\gamma^{(H)}} = O, \qquad \frac{\partial L_H(\boldsymbol{\theta}^{(H)})}{\partial \boldsymbol{\xi}_c}\Big|_{\boldsymbol{\theta}^{(H)}=\boldsymbol{\theta}_\gamma^{(H)}} = O,$$

which means $L_H$ is stationary at $\boldsymbol{\theta}_\gamma^{(H)}$ as a function of $\boldsymbol{\eta}_c$ and $\boldsymbol{\xi}_c$.

From Lemma 4, we have

$$\frac{\partial^2 L_H(\boldsymbol{\theta}^{(H)})}{\partial \boldsymbol{\xi}_c \partial \boldsymbol{\xi}_d}\Big|_{\boldsymbol{\theta}^{(H)}=\boldsymbol{\theta}_\gamma^{(H)}} = O$$

and

$$\frac{\partial^2 L_H(\boldsymbol{\theta}^{(H)})}{\partial \boldsymbol{\xi}_c \partial \boldsymbol{\eta}_d}\Big|_{\boldsymbol{\theta}^{(H)}=\boldsymbol{\theta}_\gamma^{(H)}} = (A\Lambda A^T)_{cd} \sum_{\nu: \boldsymbol{u}_{H_0*}^T \boldsymbol{x}_\nu > 0} \frac{\partial \ell_\nu(\boldsymbol{\theta}_\gamma^{(H)})}{\partial \boldsymbol{z}_\nu} \boldsymbol{x}_\nu^T.$$

Using the fact $\frac{\partial^2 \varphi(\boldsymbol{x}_\nu; \boldsymbol{u}_{H_0*})}{\partial \boldsymbol{u}_{H_0} \boldsymbol{u}_{H_0}} = 0$, we have

$$\frac{\partial^2 L_H(\boldsymbol{\theta}^{(H)})}{\partial \boldsymbol{\eta}_c \partial \boldsymbol{\eta}_d}\Big|_{\boldsymbol{\theta}^{(H)}=\boldsymbol{\theta}_\gamma^{(H)}} = O.$$

Therefore, the Hessian of $L_H$ at $\boldsymbol{\theta}_\gamma^{(H)}$ with respect to $\boldsymbol{\xi}_a, \boldsymbol{\eta}_b$ is given by

$$\begin{pmatrix} O & \tilde{F} \\ \tilde{F}^T & O \end{pmatrix}$$

where $\tilde{F} = (A\Lambda A^T) \otimes F$. Under the assumption that $F \neq O$, the eigenvalues of the above Hessian are $\{\delta_i, -\delta_i\}_{i=1}^r$, where $\{\delta_i\}_{i=1}^r$ is the singular values of $\tilde{F}$. This means there are increasing directions and decreasing directions of $L_H$ around $\boldsymbol{\theta}_\gamma^{(H)}$, and thus it is a saddle point.

## G  PAC-Bayesian bound of generalization

### G.1  Brief summary of general PAC-Bayes bound

The PAC-Bayesian framework [6, 7] has been developed for bounding generalization performance of learning models. It has been recently applied also to analysis of generalization of neural networks [8]. The following form of the bound is taken from [6].

Let $f(\boldsymbol{x}; \boldsymbol{\theta})$ be a real-valued function of $\boldsymbol{x}$ with parameter $\boldsymbol{\theta} \in \Theta$. We consider the case that the loss function $\ell(\boldsymbol{y}; \boldsymbol{z})$ is bounded, and without loss of generality assume $\ell(\boldsymbol{y}, \boldsymbol{z}) \in [0, 1]$. Training data $(\boldsymbol{x}_1, \boldsymbol{y}_1), \ldots, (\boldsymbol{x}_n, \boldsymbol{y}_n)$ is an i.i.d. sample from a distribution $\mathcal{D}$ on $(\boldsymbol{x}, \boldsymbol{y})$. Given function $f(\boldsymbol{x}; \boldsymbol{\theta})$, the training error (or empirical risk) is evaluated by

$$\hat{L}(\boldsymbol{\theta}) = \frac{1}{n} \sum_{\nu=1}^n \ell(\boldsymbol{f}(\boldsymbol{x}_\nu, \boldsymbol{\theta}), \boldsymbol{y}_\nu)$$

and the generalization error (or risk) is defined by

$$L(\boldsymbol{\theta}) = E_{\mathcal{D}}[\ell(\boldsymbol{f}(\boldsymbol{x}_\nu, \boldsymbol{\theta}), \boldsymbol{y}_\nu)].$$

In PAC-Bayes bound, we introduce a "prior" distribution $P$ on the parameter space with an assumption that $P$ does not depend on the training sample, and an arbitrary probability distribution $Q$ on $\Theta$. The distribution $Q$ may depend on the training sample. Then, for any $\delta > 0$, the inequality

$$E_Q[L(\boldsymbol{\theta})] \le E_Q[\hat{L}(\boldsymbol{\theta})] + 2\sqrt{\frac{2(KL(Q||P) + \ln\frac{n}{\delta})}{n-1}} \tag{39}$$

holds for sufficiently large $n$ with probability greater than $1 - \delta$.

First, we can see that, if the distribution of $Q$ is concentrated on a parameter set that gives very close values to $L(\hat{\boldsymbol{\theta}})$ or $\hat{L}(\hat{\boldsymbol{\theta}})$ at a parameter $\hat{\boldsymbol{\theta}}$ obtained by learning, then we have

$$E_Q[L(\boldsymbol{\theta})] \approx L(\hat{\boldsymbol{\theta}}), \qquad E_Q[\hat{L}(\boldsymbol{\theta})] \approx \hat{L}(\hat{\boldsymbol{\theta}}).$$

In such cases, Eq. (39) shows the behavior of generalization error by its upper bound involving the approximate training error and the complexity term, which is expressed by the KL-divergence.

## G.2 Generalization error bounds of embedded networks

The difference of the semi-flatness between networks of the smooth and ReLU activation can be related to the different generalization abilities of these models trough the PAC-Bayes bound Eq. (39).

### G.2.1 Choice in general cases

First we consider the general problem of choosing $P$ and $Q$ appropriately when the minimum of $\hat{L}(\boldsymbol{\theta})$ is sharp (non-flat) and can be approximated locally by a quadratic function around $\hat{\boldsymbol{\theta}}$, which is a minimum of $\hat{L}(\boldsymbol{\theta}^{(H)})$. The prior $P$ should be non-informative, and thus if $\Theta = \mathbb{R}^d$, a normal distribution $N(0, \sigma^2 I_d)$ with a large $\sigma$ is a reasonable choice. To relate the PAC-Bayes bound Eq. (39) to the generalization error at $\hat{\boldsymbol{\theta}}$, the distribution $Q$ (posterior) should distribute on parameters that do not change the empirical risk values so much from the values given by $\hat{\boldsymbol{\theta}}$. Under the assumption that $\hat{L}(\boldsymbol{\theta})$ is well approximated by a quardatic function, We set $Q$ by a normal distribution $N(0, \tau^2 \mathcal{H}^{-1})$ where $\mathcal{H}$ is the Hessian

$$\mathcal{H} := \nabla^2 \hat{L}(\hat{\boldsymbol{\theta}})$$

with a small value of $\tau$. Using the variance-covariance matrices based on the inverse Hessian is confirmed as follows. Suppose we set $Q$ by $N(\hat{\boldsymbol{\theta}}, \Sigma)$ with a general $\Sigma$ such that $\Sigma \ll \sigma^2$. Then, the Taylor series approximation of $\hat{L}(\boldsymbol{\theta}^{(H)})$ gives

$$E_Q[\hat{L}(\boldsymbol{\theta}^{(H)})] \approx \hat{L}(\hat{\boldsymbol{\theta}}^{(H)}) + \frac{1}{2}\text{Tr}[\mathcal{H}\Sigma],$$

and thus the right hand side of Eq. (39) is approximated by

$$\hat{L}(\hat{\boldsymbol{\theta}}^{(H)}) + \frac{1}{2}\text{Tr}[\mathcal{H}\Sigma] + 2\sqrt{\frac{2(KL(Q||P) + \ln\frac{n}{\delta})}{n-1}}. \tag{40}$$

It is well known that $KL(Q||P)$ with $P$ and $Q$ normal distributions is given by

$$KL(Q||P) = \frac{1}{2}\Big[\log\frac{|\sigma^2 I_d|}{|\Sigma|} + \text{Tr}[\sigma^{-2}\Sigma] + \frac{\|\hat{\boldsymbol{\theta}}\|^2}{\sigma^2} - d\Big]$$

To minimize Eq. (40) with respect to $\Sigma$, the differentiation provides the stationary condition

$$\mathcal{H} + \lambda\big(-\Sigma^{-1} + \sigma^{-2}I_d\big) = O$$

with some positive constant $\lambda$. From the assumption $\sigma^2 \gg \Sigma$, by neglecting $\sigma^{-2}I_d$, an approximate solution is given by

$$\Sigma_{opt} \approx \tau^2 \mathcal{H}^{-1},$$

where $\tau > 0$ is a scalar. Plugging this to Eq. (40) provides

$$\hat{L}(\hat{\boldsymbol{\theta}}^{(H)}) + \frac{\tau^2}{2}d + 2\sqrt{\frac{2\{d\log\frac{\sigma^2}{\tau^2} + \log\det\mathcal{H} + \frac{\tau^2}{\sigma^2}\mathrm{Tr}[\mathcal{H}^{-1}] + \frac{\|\hat{\boldsymbol{\theta}}\|^2}{\sigma^2} - d\} + 2\ln\frac{n}{\delta}}{n-1}}.$$

The second term is linear to $\tau^2$, and the main factor in the third term is $(d\log\frac{\sigma^2}{\tau^2})^{1/2}n^{-1/2}$ when $\sigma \gg 1$ and $\tau \ll 1$.

### G.2.2 The case of inactive units

We now discuss the embedding of the smooth and ReLU networks by inactive units when the training error achieves zero error. As discussed in Section 5.1, some of the parameters give flat-directions, which requires some modification of the arguments in Section G.2.1.

As notations, $\boldsymbol{\theta}_{sm}^{(H)} \in \mathbb{R}^{d_{sm}}$ and $\boldsymbol{\theta}_{rl}^{(H)} \in \mathbb{R}^{d_{rl}}$ are used for the parameters of networks with smooth and ReLU activation, respectively, and they are decomposed as $\boldsymbol{\theta}_{sm}^{(H)} = (\boldsymbol{\theta}_{sm,0}^{(H)}, \boldsymbol{\theta}_{sm,1}^{(H)}, \boldsymbol{\theta}_{sm,2}^{(H)})$ and $\boldsymbol{\theta}_{rl}^{(H)} = (\boldsymbol{\theta}_{rl,0}^{(H)}, \boldsymbol{\theta}_{rl,1}^{(H)}, \boldsymbol{\theta}_{rl,2}^{(H)})$, corresponding to the components of a copy of $\boldsymbol{\theta}^{(H_0)}$, $(\boldsymbol{v}_j)_{j=H_0+1}^{H}$, and $(\boldsymbol{w}_j)_{j=H_0+1}^{H}$. Note that the both models have the same number of surplus parameters, i.e. $\dim(\boldsymbol{\theta}_{sm,1}^{(H)}) = \dim(\boldsymbol{\theta}_{sm,2}^{(H)}) =: d_1$ and $\dim(\boldsymbol{\theta}_{rl,1}^{(H)}) = \dim(\boldsymbol{\theta}_{rl,2}^{(H)}) =: d_2$. Different choices of $P$ and $Q$ are employed in the smooth and ReLU networks: we use $P_{sm}, Q_{sm}$ for the smooth networks and $P_{rl}, Q_{rl}$ for the ReLU case.

For the smooth activation, as in Section G.2.1, a non-informative prior

$$P_{sm}: \quad N(0, \sigma^2 I)$$

is used with $\sigma \gg 1$. For the distribution $Q_{sm}$, we reflect the Hessian at the embedding by inactive units. By the definition, the directions of $(\boldsymbol{v}_j)_{j=H_0+1}^{H}$ give flat surface to $L_H$. The Hessian with respect to $(\boldsymbol{v}_j, \boldsymbol{w}_j)_{j=H_0+1}^{H}$ is thus given in the form

$$\begin{pmatrix} O & O \\ O & S \end{pmatrix},$$

where $S$ is an $(H - H_0) \times D$ dimensional symmetric matrix given by

$$S_{jk} = \sum_{\nu=1}^{n} \boldsymbol{v}_j^T \frac{\partial^2 \ell_\nu(\hat{\boldsymbol{\theta}})}{\partial \boldsymbol{z} \partial \boldsymbol{z}} \boldsymbol{v}_k \frac{\partial \varphi(\boldsymbol{x}_\nu; \boldsymbol{w}^{(0)})}{\partial \boldsymbol{w}_j} \frac{\partial \varphi(\boldsymbol{x}_\nu; \boldsymbol{w}^{(0)})}{\partial \boldsymbol{w}_k} + \delta_{jk} \sum_{\nu=1}^{n} \frac{\partial \ell_\nu(\hat{\boldsymbol{\theta}})}{\partial \boldsymbol{z}} \boldsymbol{v}_j \frac{\partial^2 \varphi(\boldsymbol{x}_\nu; \boldsymbol{w}^{(0)})}{\partial \boldsymbol{w}_j \partial \boldsymbol{w}_k}.$$

For the flat directions of $(\boldsymbol{v}_j)_{j=H_0+1}^{H}$, the same distribution as $P$ is optimal for the upper bound. Reflecting this, we set

$$Q_{sm}: \quad N(\hat{\boldsymbol{\theta}}_{sm,0}^{(H)}, \tau^2 \mathcal{H}_{sm}^{-1}) \times N(\hat{\boldsymbol{\theta}}_{sm,1}^{(H)}, \sigma^2 I_{d^1}) \times N(\hat{\boldsymbol{\theta}}_{sm,2}^{(H)}, \tau^2 S^{-1}),$$

where $\hat{\boldsymbol{\theta}}_{sm}^{(H)}$ is the embedded point and $\mathcal{H}_{sm} := \nabla^2 L_{H_0}(\boldsymbol{\theta}_{*,sm}^{(H_0)})$ is the Hessian of the narrower network.

For the ReLU networks, we first fix $K > 1$ as a constant. Since in the direction of $(\boldsymbol{w}_j)_{j=H_0+1}^{H}$ we can presume the existence of the bonded flat subset $B_K^{H-H_0}$, we define the prior $P_{rl}$ by

$$P_{rl}: \quad N(0, \sigma^2 I_{d^0}) \times N(0, \sigma^2 I_{d^1}) \times \mathrm{Unif}_{B_K^{H-H_0}}.$$

Reflecting the flat directions, the posterior $Q_{rl}$ is defined by

$$Q_{rl}: \quad N(\hat{\boldsymbol{\theta}}_{rl,0}^{(H)}, \tau^2 \mathcal{H}_{rl}^{-1}) \times N(\hat{\boldsymbol{\theta}}_{rl,1}^{(H)}, \sigma^2 I_{d^1}) \times \mathrm{Unif}_{B_K^{H-H_0}},$$

where $\mathcal{H}_{rl} := \nabla^2 L_{H_0}(\boldsymbol{\theta}_{*,rl}^{(H_0)})$ is the Hessian of the narrower network.

With these choices, the KL divergence of the smooth case is given by

$$KL(Q_{sm}||P_{sm}) = \frac{1}{2}\Big[d_{sm}^0 \log \frac{\sigma^2}{\tau^2} + d^1 \log \frac{\sigma^2}{\tau^2} + \log\det\mathcal{H}_{sm} + \log\det S$$

$$+ \mathrm{Tr}\left[\frac{\tau^2}{\sigma^2}(\mathcal{H}_{sm}^{-1} + S^{-1})\right] + \frac{\|\hat{\boldsymbol{\theta}}_{sm}\|^2}{\sigma^2} - d_{sm}^0 + d^1\Big],$$

while in the case of ReLU networks,

$$KL(Q_{rl}||P_{rl}) = \frac{1}{2}\left[d_{rl}^0 \log \frac{\sigma^2}{\tau^2} + \log \det \mathcal{H}_{rl} + \text{Tr}\left[\frac{\tau^2}{\sigma^2}\mathcal{H}_{rl}^{-1}\right] + \frac{\|\hat{\boldsymbol{\theta}}_{rl}\|^2}{\sigma^2} - d_{rl}^0\right].$$

With $\sigma \gg 1$ and $\tau \ll 1$, the major difference between these divergences comes from the term

$$d^1 \log \frac{\sigma^2}{\tau^2}$$

in the smooth networks. This suggests the advantage of the ReLU network in the overparameterized realization of zero training error in terms of the PAC-Bayesian upper bound of generalization error.

### G.2.3  The Hessian for the zero error cases

We summarize the Hessian matrix for the embedding of a global minimum that attains zero training error. For simplicity, we write only the four blocks corresponding to the surplus units.

**Smooth activation**

(I) Unit replication: As discussed in Sections 3.2 and 5.1, the the part of the Hessian is given by

$$\begin{pmatrix} O & O \\ O & \tilde{G} \end{pmatrix}. \tag{41}$$

(II) Inactive units: The part of the Hessian is given by

$$\begin{pmatrix} O & O \\ O & S_1 \end{pmatrix}, \tag{42}$$

where

$$(S_1)_{jk} = \frac{\partial^2 L_H(\hat{\boldsymbol{\theta}})}{\partial \boldsymbol{w}_j \partial \boldsymbol{w}_k} = \sum_{\nu=1}^n \boldsymbol{v}_j^T \frac{\partial^2 \ell_\nu(\hat{\boldsymbol{\theta}})}{\partial \boldsymbol{z}_\nu \partial \boldsymbol{z}_\nu} \boldsymbol{v}_k \frac{\partial \varphi(\boldsymbol{x}_\nu; \boldsymbol{w}^{(0)})}{\partial \boldsymbol{w}} \frac{\partial \varphi(\boldsymbol{x}_\nu; \boldsymbol{w}^{(0)})}{\partial \boldsymbol{w}}^T$$

$$+ \delta_{jk} \sum_{\nu=1}^n \frac{\partial \ell_\nu(\hat{\boldsymbol{\theta}})}{\partial \boldsymbol{z}_\nu} \boldsymbol{v}_j \frac{\partial^2 \varphi(\boldsymbol{x}_\nu; \boldsymbol{w}^{(0)})}{\partial \boldsymbol{w} \partial \boldsymbol{w}}.$$

(III) Inactive propagations: The part of the Hessian is given by

$$\begin{pmatrix} S_2 & O \\ O & O \end{pmatrix}, \tag{43}$$

where

$$(S_2)_{jk} = \frac{\partial^2 L_H(\hat{\boldsymbol{\theta}})}{\partial \boldsymbol{v}_j \partial \boldsymbol{v}_k} = \sum_{\nu=1}^n \frac{\partial^2 \ell_\nu(\hat{\boldsymbol{\theta}})}{\partial \boldsymbol{z}_\nu \partial \boldsymbol{z}_\nu} \varphi(\boldsymbol{x}_\nu; \boldsymbol{w}_j) \varphi(\boldsymbol{x}_\nu; \boldsymbol{w}_k).$$

We see that in all of the three cases the part of the Hessian for the surplus parameters contains a non-zero block.

**ReLU**

$(I)_R$ Unit replication: As discussed in Sections 4.2, the the part of the Hessian is given by $\begin{pmatrix} O & \tilde{F} \\ \tilde{F}^T & O \end{pmatrix}$. Since the embedded point must not be a saddle, we have $\tilde{F} = O$. As a result, the part of the Hessian is constant zero.

$(II)_R$ Inactive units: As discussed in Section 5.1, the part of the Hessian is zero.

$(III)_R$ Inactive propagations: In this case, the part of the Hessian is given by

$$\begin{pmatrix} S_3 & O \\ O & O \end{pmatrix}, \tag{44}$$

where

$$(S_2)_{jk} = \frac{\partial^2 L_H(\hat{\boldsymbol{\theta}})}{\partial \boldsymbol{v}_j \partial \boldsymbol{v}_k} = \sum_{\nu=1}^n \frac{\partial^2 \ell_\nu(\hat{\boldsymbol{\theta}})}{\partial \boldsymbol{z}_\nu \partial \boldsymbol{z}_\nu} \varphi(\boldsymbol{x}_\nu; \boldsymbol{w}_j) \varphi(\boldsymbol{x}_\nu; \boldsymbol{w}_k)$$

which is not necessarily zero unless $\varphi(\boldsymbol{x}_\nu; \boldsymbol{w}_j) = 0$ for all $\nu$.

We can see that the embedding by inactive units and unit replication give zero matrix for the part of Hessian, while the inactive propagation does not necessarily has zero matrix.