[Reviews · NeurIPS 2019]

Reviewer 1



+++ post rebuttal update: I agree with the other reviewers that the clarity of some of the definitions and results of this paper should be improved but I do not think this is a significant argument in favor of rejection. My own concerns have been addressed successfully in the rebuttal and if the authors revise their work as promised I think it is in very good shape. I will thus keep my score as it is. +++ This work generalizes the seminal Fukumizu & Amari [4] paper from 2000 in the following ways: - Section 3 studies embedding networks with size H_0 in a wider network of size arbitrary sizes H compared to just H_0+1 in [4]. Contrary to the case in [4], the networks can be of depth >3 and in this case embeddings (ii) and (iii) do no longer necessarily give rise to critical points. - Theorem 5 studies the specific 1HL case and finds that embedding (i) gives rise to a saddle point - Section 4.1 generalizes the original embeddings of [4,14] to the case of ReLU activations and shows that two of them yield more flat directions compared to smooth activation functions. - Section 4.2. shows that a minimum is always embedded as a minimum (saddle) for the inactive unit (replication) method. Again, I think this is only for 1HL networks but the Theorems do not specify this. - Finally Section 5.2 makes an argument for better generalization properties of ReLU (compared to smooth activation) networks in case of embedding a zero-residual minimizer. I am not an expert in generalization bounds so I will not comment on this matter and thus enter a lower overall confidence score. I consider these results to be interesting contributions. Yet, the generalization to deep nets is only made for the embeddings of critical points in general and not the embeddings of minima. The ReLU results are nice but at the same time not really surprising. In summary, I think the contribution is somewhat incremental but still valid and the presentation of the results is well done.

Reviewer 2



This paper studies the landscape of training loss function for neural networks. Specifically, the authors consider three methods for embedding a network into a wider one and investigate how the minima of the narrow networks perform on the landscape of the loss function for training the corresponding wide networks. The theoretical results show that the network with ReLU activation gives flatter minima which suggests better generalization performance. This paper has the following issue: 1. The contributions and results of this paper are not significant and important enough. The goal of this paper is not clear. In specific, the proposed three embedding methods can only cover a small subset of the representation functions of a wide neural network. In fact, most of the stationary points found by optimization algorithms for training wide neural networks do not correspond to the stationary points for training some small neural networks. 2. The representation of this paper is not clear. The authors propose three embedding methods including unit replication, inactive units, and inactive propagation, but do not clearly clarify them in the major theoretical results (Theorems 5 and 9). The authors should identify which embedding method is applied in these theorems and briefly discuss the corresponding theoretical results (comparison between different embedding methods). 3. For smooth activation and ReLU activation, the authors consider different embedding methods, the authors should clearly identify the difference and briefly discuss why such difference is necessary. 4. The comparison between generalization bounds for networks with ReLU activations and smooth activations is not fair because the results are derived using different choices of distributions P and Q. The authors should also discuss the generalization performance using the same choice of P and Q. 5. The experiment setting is not consistent with the theoretical results. In the experimental part, the authors set the output dimension to be 1, however, in the statement of Theorem 5, it requires that the output dimension is greater than 1. After reading rebuttal: The authors have answered my concern regarding the expressive power of the proposed embedding methods. I would like to increase my score to 5.

Reviewer 3



The paper considers three ways to make neural networks be overparameterized and their corresponding landscape analysis, including unit replication, inactive units, and inactive propagation. Both ReLU and smooth activation functions are considered. By employing PAC-Bayesian theory, the paper shows that ReLU achieves better generalization compared with Tanh. The paper is interesting. However, I have the following concerns: 1. What is the definition of "semi-flatness"? It seems not be clearly defined in main context. What is the difference of "semi-flatness" and "flatness"? 2. Employing PAC-Bayesian theory to explain the benefits of flat minima is already shown in previous literature (for example, [1]). The authors may oversell their contribution on generlization error bounds (i.e. section 5.2). [1] Neyshabur et al. Exploring generalization in deep learning. NeurIPS 2017. 3. In the paper, it seems to show that the method of unit replication method is not good since it introduces saddle points. In contrast, the method of inactive units is good since it gives the embedding semi-flat minima. How the number of added units (replicated or inactive) affects the landscape? For example, how is more inactivate (replicated) units related to the flatness (saddle point)? How does it guide through us to take advantage of specific ways of overparameterization? =======POST REBUTTAL======== I have read the rebuttal and would like to keep the score.

[Author Response · NeurIPS 2019]

We thank all the reviewers for their constructive comments. The following are our point-by-point replies.

**Rev. 1.**

**1)** *this is only for 1HL networks but the Theorems do not specify this*: The presentation in the submission might cause
confusion. While Theorems 5, 9 and 10 consider the three-layer networks for simplicity, their extension to $L$-layer
networks is easy by replacing the derivatives of the loss with the back-propagated delta. We briefly mentioned this at
the beginning of Sec. 3.2. But, we will make a clearer argument in an update.

**2)** *No definition of saddle and local minimum points*: We will include their standard definitions in Sec. 2. See also 13)
for semi-flat minima.

**3)** *Strict or non-strict saddles and what order?*: It is easy to derive the index of the saddle point based on the Hessian in
Lemma 4, but the details were skipped by space limitation. If $G$ is positive (or negative) definite and the off-diagonal $F$
is of full rank, the index is easily given by Eq.(37) in Supplements. In the other cases, the index may not be explicit,
depending on the eigenspaces of $\tilde{F}$ and $\tilde{G}$. We will add a brief comment and show the details in Supplements.

**4)** *Implications to learning with GD*: We totally agree that this is an important topic, and are currently working on it.

**5)** *Saddle by inactive units*: Theorems 2 and 4 in [4] consider a special parameter, which corresponds to $\boldsymbol{v}_{H_0} = 0$ in
the current paper. For general $\boldsymbol{v}_{H_0}$, they are not critical points, and this fact is mentioned in lines 111-112 with full
description in Supplements. If $\boldsymbol{v}_{H_0} = 0$, we can show similar results to [4]. In an update, we will clarify this at the
paragraph in line 111.

**6)** *Embedding of a saddle point*: In the unit replication, Lemma 4 tells that the Hessian of the narrower network is
recovered in the wider one. So, embedding of a saddle point with positive and negative eigenvalues gives saddle points.
For the embedding by inactive units and propagation, a critical point is not in general embedded into a critical point.

**7)** *Refer Section 2 in Introduction*: We will reflect this.

**Rev. 2.**

**8)** *Embeddings are artificial*: We do not think the three methods are artificial for the reason described in the paragraph,
line 88-93. Overparameterization is an important issue in this field as we describe in Sec. 1, and it refers to the situation
where a network has more sizes than the one necessary to realize a function. We rigorously formulate and discuss this
situation in the paper. It is also important to note that, as discussed in lines 88-93, the classical results [10,14] already
proved that the three methods of embedding are the ONLY ways of realizing a network function by a wider network in
the case of three-layer models. These embeddings are thus the essential ones in discussing overparameterization.

**9)** *Embedding methods are not clear in Theorems*: Theorem 5 does specify the embedding by $\theta_\lambda^{(H)}$, which is defined as
a symbol of the Unit Replication in Eq.(4). Additionally, Sec. 3.2 starts with a statement that we consider this type.
Theorem 9 clearly states that the embedded point is "defined by Eq(10)", which is the definition of Inactive Units (line
173). In an update, we will explicitly place the name of embedding at the theorems to avoid confusion.

**10)** *Difference of considerations for the smooth and ReLU activation*: For smooth activation, the inactive units and
propagation do not embed a critical point to a critical one in general (lines 111-112), so we discuss only the unit
replication for local minima. For ReLU, note that the definition of inactive units is different from that of smooth
activation, and discussing an embedding with inactive units is meaningful. Inactive propagation is the same as the
smooth case, and a critical point does not give a critical point in general. We will clarify this in an update.

**11)** *Different choices of P and Q in PAC-Bayes*: Using the same $P$ and $Q$ does not necessarily give a fair comparison,
but they should be chosen so that the bounds are tight. The distributions of the parameters that give similar loss values
are different, and the choice of the posterior $Q$ must reflect this difference for meaningful bounds. The prior $P$ is
arbitrary, as long as it does not depend on the training data. The choices in the paper reflect these conditions.

**12)** *Inconsistency between experiment and theory*: Sec. 5 highlights the difference between the smooth and ReLU
models, and the form Eq.(11) holds also in 1-dimensional output (see [4]), which means the difference exits also. While
the theory in Sec.5 are based on Theorem 5 or Lemma 4, the experiments use 1-dimensional network for simplicity of
realizing zero error. We will make clearer explanations on this point in the paragraph on the experiments.

**Rev. 3.**

**13)** *Definition of semi-flatness*: In Sec. 1, we say "semi-flat minima, at which a lower dimensional affine subset in the
parameter space gives a constant value of error", and use it as a definition. We will make it clearer. Flat-minima often
refer to points at which all the directions are flat. "Semi-flat" allows the case that only part of the directions are flat.

**14)** *PAC-Bayes analysis already exits*: Our motivation in Sec. 5 is to compare the smooth activation and ReLU in
overparaemterized cases. We will clarify this point better and cite the reference.

**15)** *How does the number of surplus units affect the landscape?*: Thank you for pointing out this important question.
We must admit that the detailed discussions were just left in Supplements by space limitation. For smooth networks,
some of the surplus parameters make sharp directions so the bound is linearly increasing to the surplus number of
units, while in ReLU there are no sharp directions so that no such major increasing terms exist in the bounds. This is
illustrated in the experimental results (Fig.2(b)). We will emphasize this in an update.

[Meta-Review · NeurIPS 2019]

The original scores assigned by the reviewers had a large variance. Some of the problems raised by the reviewers were however addressed in the rebuttal and one reviewer raised his score. After the discussion period, the reviewers agree that the paper makes some novel contributions, arguably novel for some reviewers. They also agree that the clarity of the paper should be improved. Overall, the topic addressed is an important one and it seems that the authors could improve the presentation for the final version, justifying acceptance as a poster.